# Marcks and Marcks-like 1 proteins promote spinal cord development and regeneration in *Xenopus*

**Mohamed El Amri[1,2], Abhay Pandit[2], Gerhard Schlosser[1]\***

[1]School of Biological and Chemical Sciences, University of Galway, Galway, Ireland; [2]Research Ireland Center for Medical Devices (CÚRAM), University of Galway, Galway, Ireland

**Abstract** Marcks and Marcksl1 are abundant proteins that shuttle between the cytoplasm and membrane to modulate multiple cellular processes, including cytoskeletal dynamics, proliferation, and secretion. Here, we performed loss- and gain-of-function experiments in *Xenopus laevis* to reveal the novel roles of these proteins in spinal cord development and regeneration. We show that Marcks and Marcksl1 have partly redundant functions and are required for normal neurite formation and proliferation of neuro-glial progenitors during embryonic spinal cord development and for its regeneration during tadpole stages. Rescue experiments in Marcks and Marcksl1 loss-of-function animals further suggested that some of the functions of Marcks and Marcksl1 in the spinal cord are mediated by phospholipid signaling. Taken together, these findings identify Marcks and Marcksl1 as critical new players in spinal cord development and regeneration and suggest new pathways to be targeted for therapeutic stimulation of spinal cord regeneration in human patients.

## Editor's evaluation

This important work addresses the role of Marcks/Markcksl during spinal cord development and regeneration. The study is exceptional in combining molecular approaches to understand the mechanisms of tissue regeneration with behavioural assays, which is not commonly employed in the field. The data presented is convincing and comprehensive, using many complementary methodologies.

**\*For correspondence:** gerhard.schlosser@ universityofgalway.ie

**Competing interest:** The authors declare that no competing interests exist.

## Introduction

Spinal cord injury (SCI) in humans is a life-changing condition with no effective treatment. However, many non-mammalian vertebrates can fully regenerate their spinal cords after injury (***Lee-Liu et al., 2013***; ***Alunni and Bally-Cuif, 2016***; ***Rodemer et al., 2020***). In the clawed toad *Xenopus*, tadpoles can fully regenerate the spinal cord; however, this capacity is lost during metamorphosis (***Beattie et al., 1990***; ***Gibbs et al., 2011***). Therefore, *Xenopus* offers a rare opportunity to compare cellular and molecular responses to SCI in a regeneration-competent system with a regeneration-incompetent system in the same species. Thus, differences between the responses to SCI in the two stages likely have some functional implications for the difference in regenerative capacity rather than reflecting other phylogenetically accumulated differences in development, as may be the case when different species are compared.

A recent proteomic screen to identify proteins differentially regulated in regeneration-competent tadpoles versus regeneration-incompetent froglets after SCI identified a number of proteins that are highly upregulated at the tadpole stage but downregulated in froglets (***Kshirsagar, 2020***). These proteins included Marcks and Marcksl1, which had previously been implicated in the regeneration of

other tissues (*El Amri et al., 2018*) suggesting a potential role for these proteins also in spinal cord regeneration.

Marcks is an abundant and widely expressed protein with three highly conserved functional domains: an effector domain (ED), a Marcks Homology 2 (MH2) domain, and a myristoylation domain (*Brudvig and Weimer, 2015*; *El Amri et al., 2018*). Marcks-like protein 1 (Marcksl1) shares strong homology and functionality with Marcks (*Stumpo et al., 1998*). In an unphosphorylated state, the positively charged ED of Marcks attaches to the plasma membrane, and the myristoylation site tethers the protein to the membrane (*Kim et al., 1994*; *McLaughlin and Aderem, 1995*). However, when the ED is phosphorylated by protein kinase C (PKC), thereby acquiring more negative charges, Marcks translocates to the cytoplasm (*Brudvig and Weimer, 2015*; *McLaughlin and Aderem, 1995*).

Depending on their phosphorylation status and subcellular localization, Marcks and Marcksl1 regulate multiple processes in the cell. For example, they have been shown to modulate the polymerization and cross-linking of actin filaments, recruit secretory vesicles to the membrane, and interact with phosphatidylinositol 4,5-bisphosphate ($PIP_2$), thereby regulating multiple signaling pathways downstream of $PIP_2$ such as phospholipase C (PLC) and phospholipase D (PLD) dependent pathways (*Hartwig et al., 1992*; *Morash et al., 1998*; *Yarmola et al., 2001*; *Wohnsland et al., 2000*; *McLaughlin et al., 2002*; *Sundaram et al., 2004*; *Xu et al., 2014*). Consequently, Marcks and Marcksl1 affect cytoskeletal dynamics, cell migration, secretion, cell proliferation, and differentiation in the nervous system and other tissues (*Brudvig and Weimer, 2015*; *El Amri et al., 2018*).

Importantly, Marcks and Marcksl1 proteins were found to be upregulated in several different regenerative processes, such as lens regeneration in newts, fin regeneration in lungfish and bichirs, tail regeneration in lizards, and cardiac tissue regeneration following infarction in mice (*Bock-Marquette et al., 2009*; *Sousounis et al., 2014*; *Nogueira et al., 2016*; *Alibardi, 2017*; *Lu et al., 2019*). Moreover, a Marcks-like protein in axolotls (AxMLP) initiates an early proliferative response in axolotl tail and limb regeneration (*Sugiura et al., 2016*). In the nervous system, Marcks and Marcksl1 are also significantly upregulated in outgrowing and/or regenerating neurites of facial motor neurons, dorsal root, and superior cervical ganglia, in the optic nerve and during axonal sprouting after brain stroke, suggesting an essential role for Marcks and Marcksl1 in neurite outgrowth and regeneration in both the peripheral and central nervous systems (*McNamara et al., 2000*; *Carmichael et al., 2005*; *Szpara et al., 2007*; *Veldman et al., 2007*).

In this study, we combined Marcks and Marcksl1 gain- and loss-of-function experiments in *Xenopus laevis* with histological and behavioral analyses to show, for the first time, that Marcks and Marcksl1 also play essential roles during spinal cord development and regeneration. We demonstrate that both proteins have partly redundant functions and are required for normal neurite outgrowth and the proliferation of neuro-glial progenitors during the development of the spinal cord, as well as for its regeneration after full transection. Rescue experiments with pharmacological compounds in Marcks and Marcksl1 loss-of-function animals further suggest that at least some of the functions of Marcks and Marcksl1 in the spinal cord are mediated by phospholipid signaling. Taken together, these findings identify Marcks and Marcksl1 as critical new players in spinal cord development and regeneration and highlight the power of *Xenopus* as an experimental model for identifying new pathways required for regeneration competence.

## Results

### Marcks and Marcksl1 are expressed in the developing *Xenopus* spinal cord

To elucidate the role of Marcks and Marcksl1 for spinal cord development, we first analyzed mRNA expression of *marcks.S*, *marcksl1.S*, and *marcksl1.L* in embryos of *X. laevis* at stages 26 and 34 (early and late tailbud stage, respectively). Due to the 94% sequence identity between *marcks.S* and *marcks.L* mRNAs, the *marcks.S* probe is expected to cross-react with *marcks.L*, while *marcksl1.S* and *marcksl1.L* mRNA have only 84% identity and show slightly different expression suggesting that they exhibit only limited if any cross-reactivity. Whole-mount in situ hybridization showed that all of these genes have a similar pattern of expression (*Figure 1A–G*, *Figure 1—figure supplement 1*). They are expressed in the brain and spinal cord as well as in the pharyngeal arches, somites, intermediate mesoderm, and blood islands of the ventrolateral mesoderm confirming previous reports (*Gawantka et al., 1998*;

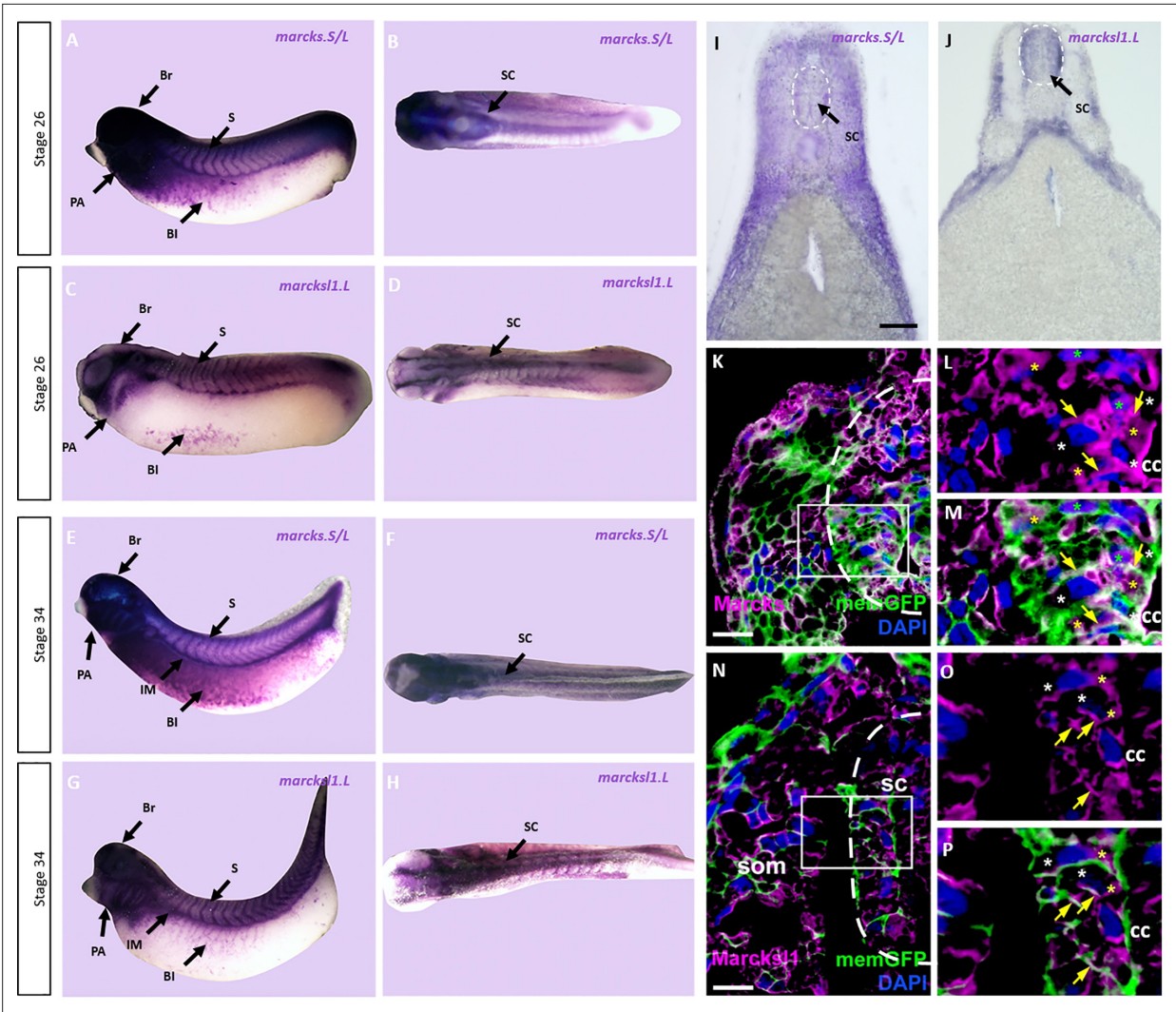

**Figure 1.** Expression of Marcks and Marcksl1 in the *Xenopus* spinal cord. Expression of *marcks.S* (**A, B, E, F**) and *marcksl1.L* (**C, D, G, H**) as revealed by whole-mount in situ hybridization in stage 26 (**A–D**) and 34 (**E–H**) *Xenopus* embryos in lateral (**A, C, E, G**) and dorsal (**B, D, F, H**) views (anterior is to the left). Expression in the spinal cord (SC), brain (Br), somites (S), intermediate mesoderm (IM), blood islands (Bl), and pharyngeal arches (PA) is indicated. Transverse sections showing expression of *marcks.S* (**I**) and *marcksl1.L* (**J**) in the spinal cord at stage 34. Immunostaining for Marcks (**K–M**) and Marcksl1 (**N–P**) protein in confocal images of transverse sections through stage 26 embryos after *memGFP* mRNA injection. White boxed areas in (**K**) and (**N**) are shown at a higher magnification in (**L, M**) and (**O, P**), respectively. Yellow asterisks indicate the expression of Marcks in the cytoplasm; green asterisks indicate expression in nuclei; white asterisks show the absence of expression in the cytoplasm; yellow arrows show expression in the membrane. Spinal cord outlined with hatched line in (**I, J, K, N**). Abbreviations: cc, central canal; sc, spinal cord; som, somites. Scale bar in **I** = 50 µm for (**I, J**). Scale bar in **K** and **N** = 20 µm.

The online version of this article includes the following figure supplement(s) for figure 1:

**Figure supplement 1.** Expression of *marcksl1.S* in *Xenopus*.

**Figure supplement 2.** Expression of *marcks.S* mRNA revealed by transverse vibratome sections of *Xenopus* stage 34 embryos after whole-mount in situ hybridization.

**Figure supplement 3.** Expression of *marcksl1.L* mRNA as revealed by transverse vibratome sections of *Xenopus* stage 34 embryos after whole-mount in situ hybridization.

**Figure supplement 4.** Expression of *marcksl1.S* mRNA as revealed by transverse vibratome sections of *Xenopus* stage 34 embryos after whole-mount in situ hybridization.

**Figure supplement 5.** Specificity of Marcks and Marcksl1 antibodies.

**Figure supplement 6.** Distribution of Marcks and Marcksl1 proteins in the *Xenopus* spinal cord.

*Zhao et al., 2001*). *marcks.S* is more broadly distributed through the entire brain, whereas *marcksl1.L* and *marcksl1.S* have a more restricted expression and are absent from the forebrain. Vibratome sections of stage 34 embryos after whole-mount in situ hybridization confirmed the expression of all genes throughout the spinal cord as well as in the brain, lens, pharyngeal arches, somites, and other mesodermal derivatives (*Figure 1H, I*; *Figure 1—figure supplements 2–4*). They also reveal the expression of all genes in the epidermis and of *marcks.S* in the otic vesicle and show that *marcksl1.S* has a weaker and more restricted expression in the brain and spinal cord than *marcksl1.L*.

We next used two polyclonal antibodies raised against epitopes shared between *X. laevis* Marcks.S and Marcks.L protein and between Marcksl1.S and Marcksl1.L, respectively, to clarify the subcellular distribution of Marcks and Marcksl1 proteins in the developing spinal cord. These antibodies specifically recognize Marcks and Marcksl1 proteins, respectively, without cross-reactivity (*Figure 1—figure supplement 5*). Confocal microscopy of embryos injected with mRNAs encoding a membrane-tethered form of GFP (*memGFP*) and transversely sectioned at stage 26 revealed that Marcks and Marcksl1 are mostly associated with outer cell membranes. This is true for the ventricular zone, where progenitor cells are located, and the adjacent mantle zone, which harbors migrating and differentiating neurons (*Figure 1K–P*, *Figure 1—figure supplement 6*). In addition, a subset of cells in both ventricular and mantle zones shows cytoplasmic expression of Marcks and Marcksl1. Marcks, but not Marcksl1 was also occasionally seen in cell nuclei.

Because shuttling between the membrane and cytoplasm is essential for the function of Marcks and Marcksl1 in other developmental contexts (*El Amri et al., 2018*), their subcellular localization to membranes and cytoplasm in the spinal cord and their widespread distribution in the developing spinal cord suggests that these proteins may play hitherto unrecognized roles during normal spinal cord development. To test this hypothesis, we next used two independent approaches, injection of antisense Morpholino oligonucleotides (MOs) and CRISPR/Cas9-mediated gene editing, to determine how loss of function of these proteins affects neurite outgrowth, cell proliferation, and neuro-glial progenitor populations during spinal cord development.

## Marcks and Marcksl1 are required for normal neurite formation during spinal cord development

We first studied the role of Marcks and Marcksl1 in neurite formation in the spinal cord individually by injecting a single blastomere in 4- to 8-cell stage embryos with either a MO blocking the translation of both *marcks.L* and *marcks.S* or with a mixture of two MOs blocking the translation of *marcksl1.L* and *marcksl1.S*, respectively. To minimize the gastrulation and axis defects previously observed after knockdown of *marcks* or *marcksl1* (*Zhao et al., 2001*; *Iioka et al., 2004*), we injected lower amounts of MOs (9 ng) than in previous studies (16–40 ng). This resulted in the unilateral knockdown of the respective proteins on the injected side of the embryo, whereas the uninjected side developed normally. *myc-GFP* mRNA was co-injected to identify the injected side. The effectiveness and specificity of all MOs used were validated by in vitro transcription and translation of the various proteins in the presence or absence of the appropriate MO (*Figure 2—figure supplement 1*). None of these injections led to significant alterations in immunostaining for the neurite marker acetylated tubulin on the injected side of the spinal cord relative to the uninjected side in stage 34 embryos compared to an unspecific control MO (*Figure 2A–C*). However, when the three MOs targeting *marcks.S/L*, *marcksl1.S*, and *marcksl1.L* were co-injected, thus blocking all four homeologs (4M MO), acetylated tubulin staining was significantly decreased on the injected side (*Figure 2D, E, G*). Co-injection of mRNA constructs encoding chicken Marcks and *Xenopus tropicalis* Marcksl1, which were selectively mutated to prevent MO binding, resulted in complete rescue of acetylated tubulin staining, confirming the specific action of the MOs used (*Figure 2F, G*). This suggests that Marcks and Marcksl1 are required for neurite formation during spinal cord development but act in a largely redundant fashion. To account for these overlapping functions, we aimed to simultaneously knockdown or gene edit all four proteins (Marcks.L/S, Marcksl1.L/S) for the rest of our study.

To verify the results of our MO knockdown experiments, we analyzed F0 embryos after CRISPR–Cas9 gene editing. For these experiments, single-guide RNA constructs targeting *marcks.S*, *marcks.L*, *marcksl1.S*, and *marcksl1.L* genes were co-injected with Cas9 as a ribonucleoprotein (RNP) and *myc-GFP* mRNA into one blastomere during the 1-cell stage for genotyping or the 4- to 8-cell stage for phenotypic analysis. At later embryonic stages (stages 26–42), genomic DNA was extracted from

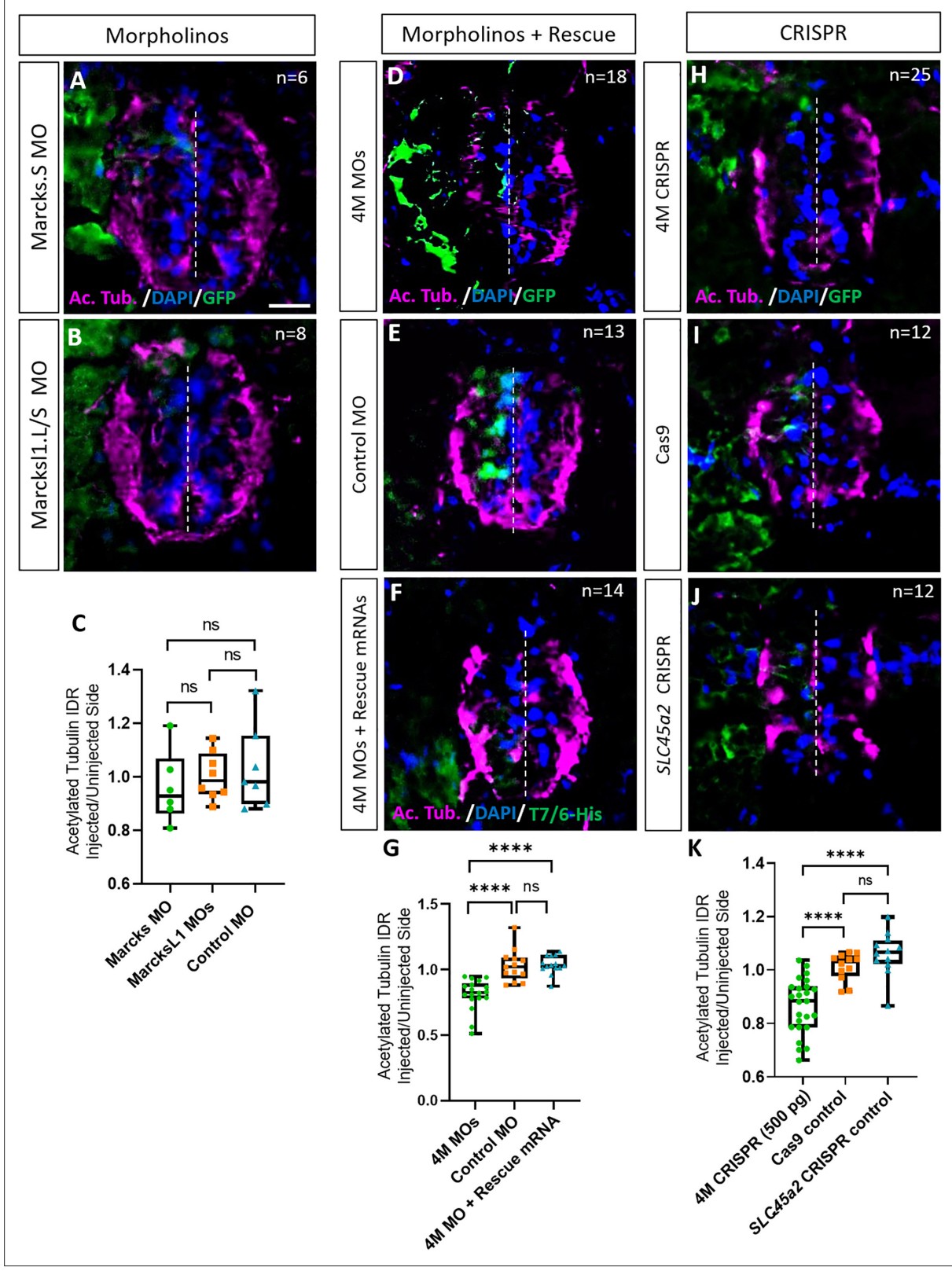

**Figure 2.** Marcks and Marcksl1 are required for neurite outgrowth during *Xenopus* spinal cord development. (**A–C**) Evaluation of neurite outgrowth after individual morpholino (MO) knockdown of Marcks or Marcksl1 in stage 34–36 developing spinal cords. A dorsal blastomere of 4- to 8-cell stage embryos was injected with either 9 ng of marcks MOs (**A, C**), 9 ng of marcksl1.L and marcksl1.S MOs each (**B, C**), or 9 ng of control MO (**C**). All injections contained 125 pg of *myc-GFP* mRNA to identify the injected side. (**D–J**) Evaluation of neurite outgrowth after loss of function of all four homeologs

*Figure 2 continued on next page*

*Figure 2 continued*

(*marcks.S, marcks.L, marcksl1.S*, and *marcksl1.L*) by MO knockdown (4M MO) or CRISPR gene editing (4M CRISPR). A dorsal blastomere of 4- to 8-cell stage embryos was co-injected with 9 ng of each MO (marcks.S, marcksl1.L, and marcksl1.S MO) (**D, G**); 27 ng standard control MO (**E, G**); 9 ng of each MO plus 300 pg of mutated chicken *marcks* and 350 pg of mutated *X. tropicalis marcksl1* rescue mRNA constructs encoding T7 and 6-His tags (**F, G**); 500 pg of each single-guide RNA (*marcks.L/S* and *marcksl1.L/S* sgRNA) with 1 ng Cas9 protein (**H, K**); 1 ng Cas9 protein (**I, K**); or *slc45a2* sgRNA and 1 ng Cas9 protein (**J, K**). All injections contained 125 pg of myc-GFP mRNA to identify the injected side. Injected (left) and uninjected (right) spinal cord sides (separated by the dashed line) were compared in transverse sections immunostained for acetylated tubulin (**A, B, D–F, H–J**) and for GFP, T7, and 6-His to determine the injected side. Immunostaining for acetylated tubulin was quantified (**G, K**) by determining the integrated density ratio (IDR) of staining in injected to uninjected sides. Significance was determined using an ordinary one-way ANOVA with Tukey's multiple comparisons test. NS, not significant; ****$p < 0.0001$. Error bars represent minimum to maximum values showing all points with a line at the median. *n* refers to the number of animals analyzed. Scale bar in **A** = 25 µm (for all panels).

The online version of this article includes the following source data and figure supplement(s) for figure 2:

**Figure supplement 1.** Validation of Marcks and Marcksl1 knockdown and gene editing.

**Figure supplement 1—source data 1.** Full size gel pictures for *Figure 2A* (labeled).

**Figure supplement 1—source data 2.** Full size gel pictures for *Figure 2A* (unlabeled).

**Figure supplement 2.** Control *slc45a2* sgRNA/Cas9 efficiently and specifically edits targeted *X. laevis* DNA.

**Figure supplement 3.** *marcks.L/S* sgRNA efficiently and specifically edits *Xenopus laevis marcks.L* DNA.

**Figure supplement 4.** *marcks.L/S* sgRNA efficiently and specifically edits *Xenopus laevis marcks.S* DNA.

**Figure supplement 5.** *marcksl1.L/S* sgRNA efficiently and specifically edits *X. laevis marcksl1.L* DNA.

**Figure supplement 6.** *marcksl1.L/S* sgRNA efficiently and specifically edits *X. laevis marcksl1.S* DNA.

**Figure supplement 7.** Marcks and Marcksl1 overexpression promotes neurite outgrowth during *Xenopus* spinal cord development.

---

some of the embryos, sequenced, and analyzed with Synthego Inference for CRISPR Edits (ICE) to determine the frequency with which indels were generated in the targeted region (indel score) as well as the frequency of indels with frameshifts (knockout score). Sibling embryos were used for phenotypic analysis. In control experiments, we either injected Cas9 protein and *myc-GFP* mRNA alone or injected a sgRNA targeting *slc45a2* (*DeLay et al., 2018*), encoding a protein that mediates melanin synthesis, Cas9 proteins, and *myc-GFP* mRNA. When analyzing CRISPR/Cas9-edited animals in the F0 generation, a considerable and variable extent of mosaicism has to be taken into account because different gene editing events may occur in different subsets of cells (*Blitz et al., 2013*; *DeLay et al., 2018*). We, therefore, first tested a range of different sgRNAs to ensure high overall gene editing and knockout efficiencies.

We first validated our CRISPR knockout procedure using the control gene *slc45a2*. Embryos injected during the 1-cell stage with *slc45a2* RNPs and analyzed at stages 37–46 lost pigmentation in both eyes, whereas embryos injected during the 2- to 4-cell stage showed loss of pigmentation only in the injected side. Confirming a previous report (*DeLay et al., 2018*), genomic sequencing revealed a high indel score (93%) and knockout score (67–68%) in these embryos (*Figure 2—figure supplement 2*). Almost all surviving embryos displayed visible oculocutaneous albinism, with the majority showing almost no eye pigment (*Figure 2—figure supplement 2*).

Next, we designed, synthesized, and tested a total of 18 sgRNA transcripts targeting either *marcks.L/S* or *marcksl1.L/S* (sgRNAs were designed to target both homeologs of each gene) and selected the sgRNA transcripts creating the highest and most consistent number of indels and knockout scores for each pair of genes – named *marcks.L/S* and *marcksl1.L/S* sgRNA, respectively – for further experiments (*Figure 2—figure supplements 3–6*). The *marcks.L/S* sgRNA introduced indels in 88% of the targeted DNA from *marcks.L* exon 1 (8 embryos) and 84% from *marcks.S* exon 1 (8 embryos), with an average of 78% and 76%, respectively, consisting of frameshift mutations which may contribute to a functional knockout (*Figure 2—figure supplement 3C and 4C*). The *marcksl1.L/S* sgRNA introduced indels in 73% of the targeted DNA from *marcksl1.L* exon 2 (six embryos) and 73% from *marcksl1.S* exon 2 (seven embryos), with an average of 50% and 47%, respectively, being out-of-frame (*Figure 2—figure supplement 5C and 6C*). We also confirmed by immunohistochemistry that co-injection of *marcks.L/S* and *marcksl1.L/S* sgRNA, which is predicted to edit all four homeologs (henceforth denoted as 4M CRISPR) drastically reduced immunostaining for Marcks and Marcksl1 protein on the injected side (*Figure 2—figure supplement 1B–G*), indicating that protein levels are reduced in gene-edited embryos.

Similar to the knockdown of the proteins by MOs, the unilateral CRISPR-mediated knockout of all Marcks and Marcksl1 homeologs (4M CRISPR) led to a significant reduction in acetylated tubulin staining on the injected side compared to Cas9 and *slc45a2* controls (*Figure 2H–K*). Conversely, over-expression of both proteins by injection of *marcks* and *marcksl1* mRNA at two different doses (100 or 300 pg each) together with *myc-GFP* mRNA resulted in a significant increase in acetylated tubulin staining on the injected side compared to that in control embryos injected with *myc-GFP* mRNA alone (*Figure 2—figure supplement 7*). Taken together, these results suggest that Marcks and Marcksl1 play essential roles for neurite formation during normal embryonic development of the *Xenopus* spinal cord, with elevated levels of Marcks and Marcksl1 being able to further promote neurite outgrowth.

## Marcks and Marcksl1 are required for the proliferation of neuro-glial progenitors during spinal cord development

Next, we tested whether Marcks and Marcksl1 are required for the proliferation of neuro-glial progenitor cells during spinal cord development. After unilateral injection with a cocktail of the three MOs targeting *marcks.S/L*, *marcksl1.S* and *marcksl1.L* (4M MO) or with the combined *marcks.L/S* and *marcksl1.L/S* sgRNAs (4M CRISPR), we analyzed the number of cells in mitosis in the spinal cord of stage 34 embryos by immunostaining against phospho-histone H-3 (PH3). In both control and experimental animals, PH3-positive nuclei are confined to the ventricular zone at this stage, where neuro-glial progenitors reside (*Figure 3A–C, E–G*). The proportion of PH3-positive ventricular nuclei on the injected side (expressed as the ratio between injected/uninjected sides) was significantly reduced in both experiments compared to control injections (*Figure 3A–H*). Conversely, the injection of *marcks* and *marcksl1* mRNA together with *myc-GFP* mRNA resulted in a significant increase in the proportion of PH3-positive ventricular nuclei on the injected side compared to control embryos injected with *myc-GFP* mRNA alone (*Figure 3—figure supplement 1*). These observations indicate that Marcks and Marcksl1 are required for normal patterns of cell proliferation in the ventricular zone of the developing spinal cord. In addition, increased levels of Marcks and Marcksl1 are sufficient to further promote cell proliferation.

To directly test the role of Marcks and Marcksl1 for neuro-glial progenitors, we next immunostained spinal cords of stage 34 embryos for the progenitor markers Sox2 or Sox3 after unilateral injection of *marcks.L/S* and *marcksl1.L/S* sgRNAs. The number of Sox2- or Sox3-positive nuclei on the injected side (expressed as a ratio between injected/uninjected sides) was significantly reduced as compared to control embryos injected with *slc45a2* sgRNAs or with Cas9 alone (*Figure 3I–M*). This indicates that Marcks and Marcksl1 are required for maintaining a normal pool of neuro-glial progenitor cells during spinal cord development. This may at least partly result from the role of both proteins in promoting the proliferation of these neuro-glial progenitors, as described above, but additional roles for the survival and maintenance of neuro-glial progenitors cannot be ruled out.

In spite of the reduction in progenitor proliferation after Marcks/Marcksl1 loss of function, overall cell numbers (as indicated by the numbers of DAPI+ nuclei) on the injected side were not yet significantly reduced in these embryos (*Figure 3N, O*) indicating that the decrease in neurite staining described in the previous section cannot be attributed to a reduction in overall cell number.

## Phospholipid signaling mediates the functions of Marcks and Marcksl1 during spinal cord development

Since Marcks and Marcksl1 are known to be multifunctional proteins, we next asked which signaling pathways may mediate the effects of Marcks and Marcksl1 during spinal cord development. Our experiments described so far suggested essential functions of these proteins in both proliferation control and neurite outgrowth. We, therefore, focused on phospholipid signaling since previous studies have suggested that the main phospholipid signaling pathways, which depend on PLC and PLD, are regulated by Marcks (*Glaser et al., 1996*; *McLaughlin et al., 2002*; *Morash et al., 2000*; *Morash et al., 1998*) and have essential functions in both the regulation of mitosis and of cytoskeletal dynamics including neurite outgrowth (*Rebecchi and Pentyala, 2000*; *Saarikangas et al., 2010*; *Kam and Exton, 2001*; *Foster and Xu, 2003*; *Zhang et al., 2004*),. To determine whether the essential roles of Marcks and Marcksl1 for neurite formation and proliferation of neuronal progenitors are mediated by phospholipid signaling, we attempted to rescue the phenotype of 4M-CRISPants with

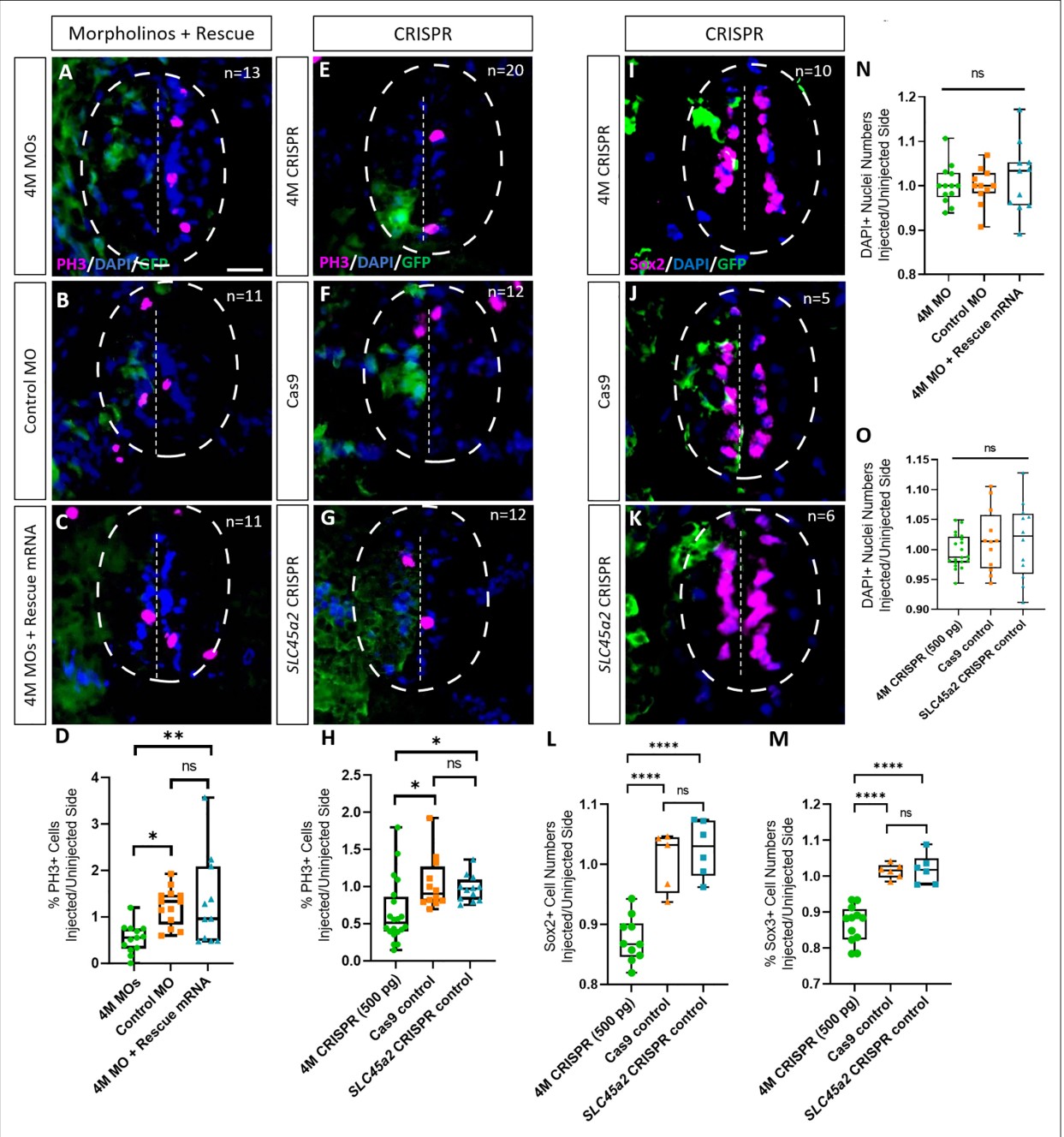

**Figure 3.** Marcks and Marcksl1 are required for the proliferation of neuro-glial progenitors during *Xenopus* spinal cord development. Evaluation of mitotic activity (**A–H**), number of neuro-glial progenitors (**I–M**), and total cell numbers (**N, O**) after loss of function of all four homeologs (*marcks.S, marcks.L, marcksl1.S,* and *marcksl1.L*) by MO knockdown (4M MO) or CRISPR gene editing (4M CRISPR). A dorsal blastomere of 4- to 8-cell stage embryos was co-injected with 9 ng of each MO (marcks.S, marcksl1.L, and marcksl1.S MO) (**A, D**); 27 ng standard control MO (**B, D**); 9 ng of each MO plus 300 pg of mutated chicken *marcks* and 350 pg of mutated *X. tropicalis marcksl1* rescue mRNA constructs encoding T7 and 6-His tags (**C, D, N**); 500 pg of each single-guide RNA (*marcks.L/S* and *marcksl1.L/S* sgRNA) with 1 ng Cas9 protein (**E, H, I**); 1 ng Cas9 protein (**F, H, J, L, M, O**); or *slc45a2* sgRNA and 1 ng Cas9 protein (**G, H, K, L, M, O**). All injections contained 125 pg of *myc-GFP* mRNA to identify the injected side. Injected (left) and uninjected (right) spinal cord sides (separated by the dashed line; spinal cord outlined with hatched line) were compared in transverse sections of stage 34–36 embryos immunostained for phospho-histone 3 (PH3) (**A–C, E–G**) or Sox2 (**I–K**) and for GFP, T7, and 6-His to determine the injected side. Mitotic or neuro-glial progenitor cells were quantified by determining the ratio of the percentage of PH3+ cells (**D, H**) and the numbers of Sox2+ (**L**) or Sox3+ (**M**) cells on injected and uninjected sides. DAPI+ nuclei were counted to quantify overall cell numbers (**N, O**). Significance was determined using an ordinary one-way ANOVA with Tukey's multiple comparisons test. NS, not significant; *p < 0.05; **p ≤ 0.01; ****p < 0.0001. Error bars represent minimum to maximum values showing all points with the line at median. *n* refers to the number of animals analyzed. Scale bar in **A** = 25 μm (for all panels).

*Figure 3 continued on next page*

**Figure supplement 1.** Marcks and Marcksl1 overexpression promotes mitotic activity during *Xenopus* spinal cord development.

Sphingosine-1-phosphate (S1P) dissolved in 2% dimethyl sulfoxide (DMSO), which activates both PLC and PLD signaling (*Meacci et al., 2003*; *Siehler and Manning, 2002*).

For these and subsequent pharmacological experiments, embryos were unilaterally injected with *marcks.L/S* and *marcksl1.L/S* sgRNAs (4M CRISPR), as described above, were then raised for 1 day before fixation in the pharmacological compound and were analyzed at stage 34 by immunostaining for the neurite marker acetylated tubulin and the mitosis marker PH3. 4M CRISPR-injected control embryos were raised in 2% DMSO and similarly analyzed. Because pharmacological compounds may affect the spinal cord in multiple ways, not all of which may be Marcks/Marcksl1 dependent, we present our data here as ratios of acetylated tubulin or PH3 staining in injected/uninjected sides (*Figure 4*). This allows us to specifically determine, whether compounds are able to rescue deficiencies in Marcks/Marcksl1-depleted spinal cords independent of their potentially additional effects on spinal cord development. For completeness, the individual effects of pharmacological treatments on the injected and uninjected sides of embryos are shown in *Figure 4—figure supplement 1*.

Compared to DMSO-treated controls, incubation in the phospholipid signaling activator S1P significantly increased the intensity of acetylated tubulin staining (*Figure 4A*) as well as the proportion of PH3-positive cells (*Figure 4B*) on the CRISPR-injected side relative to the uninjected side suggesting that Marcks and Marcksl1 promote neurite outgrowth and progenitor proliferation at least partly by a phospholipid-dependent mechanism. However, after incubation in the compound M-3M3FBS, which specifically activates PLC-dependent phospholipid signaling (*Bae et al., 2003*), only the proportion of PH3-positive cells was significantly increased compared to the controls (*Figure 4D*), but acetylated tubulin was not (*Figure 4C*), suggesting that PLC signaling may mediate some of the effects of Marcks and Marcksl1 on progenitor proliferation but not on neurite formation. In contrast, neither the specific PLD inhibitor FIPI (*Monovich et al., 2007*; *Su et al., 2009*), nor the specific PLC inhibitor U-73122 (*Bleasdale et al., 1989*) had any significant effect on neurite formation or progenitor proliferation in Marcks/Marcksl1-depleted spinal cords (*Figure 4A–D*).

Because phospholipid signaling downstream of Marcks has previously been suggested to be $PIP_2$ dependent (*Morash et al., 2000*; *Sundaram et al., 2004*), we next tested whether enhancement or inhibition of $PIP_2$ synthesis is also able to modulate the effects on spinal cord development in our CRISPR-injected embryos. For $PIP_2$ biosynthesis enhancement and inhibition, we incubated 4M CRISPR-injected embryos in *N*-methyllidocaine iodide (NMI) and ISA-2011B, respectively (*Lee et al., 1995*; *Semenas et al., 2014*). Compared to DMSO-treated controls, embryos in which NMI enhanced $PIP_2$ biosynthesis showed a significant increase in the intensity of acetylated tubulin staining (*Figure 4E*) as well as in the proportion of PH3-positive cells (*Figure 4F*) on the CRISPR-injected side relative to the uninjected side. In contrast, the $PIP_2$ synthesis inhibitor ISA-2011B had no significant effect (*Figure 4E, F*). This indicates that the stimulation of neurite formation and progenitor proliferation by Marcks and Marcksl1 is indeed $PIP_2$ dependent. While $PIP_2$ is generally thought to act upstream of PLC and PLD, additional experiments will be needed to confirm this for the developing spinal cord.

The ability of Marcks to promote $PIP_2$ signaling has been previously suggested to be linked to the subcellular localization of the protein, which depends on its phosphorylation status, with unphosphorylated Marcks being mostly membrane-bound and phosphorylated Marcks being largely cytoplasmic (*Kim et al., 1994*; *McLaughlin and Aderem, 1995*). In turn, the phosphorylation status of Marcks is regulated by PKC (*McLaughlin and Aderem, 1995*). Although Marcks and Marcksl1 proteins are severely depleted in our CRISPR-injected embryos, clones of cells with at least partially functional Marcks and Marcksl1 proteins are expected to be present due to the mosaicism of CRISPR gene editing effects in our F0 embryos. Assuming that subcellular localization of Marcks and Marcksl1 is functionally important in the spinal cord, we should thus be able to partially rescue the deficiencies of our CRISPR embryos by modulating PKC activity.

To test this hypothesis, we used the potent PKC activator Phorbol 12-myristate 13-acetate (PMA) and the broad-spectrum PKC inhibitor Go6983 (*Liu and Heckman, 1998*; *Shen, 2003*). Compared to DMSO-treated controls, embryos treated with the PKC inhibitor Go6983 showed a significant increase in the intensity of acetylated tubulin staining (*Figure 4G*) as well as in the proportion of PH3-positive

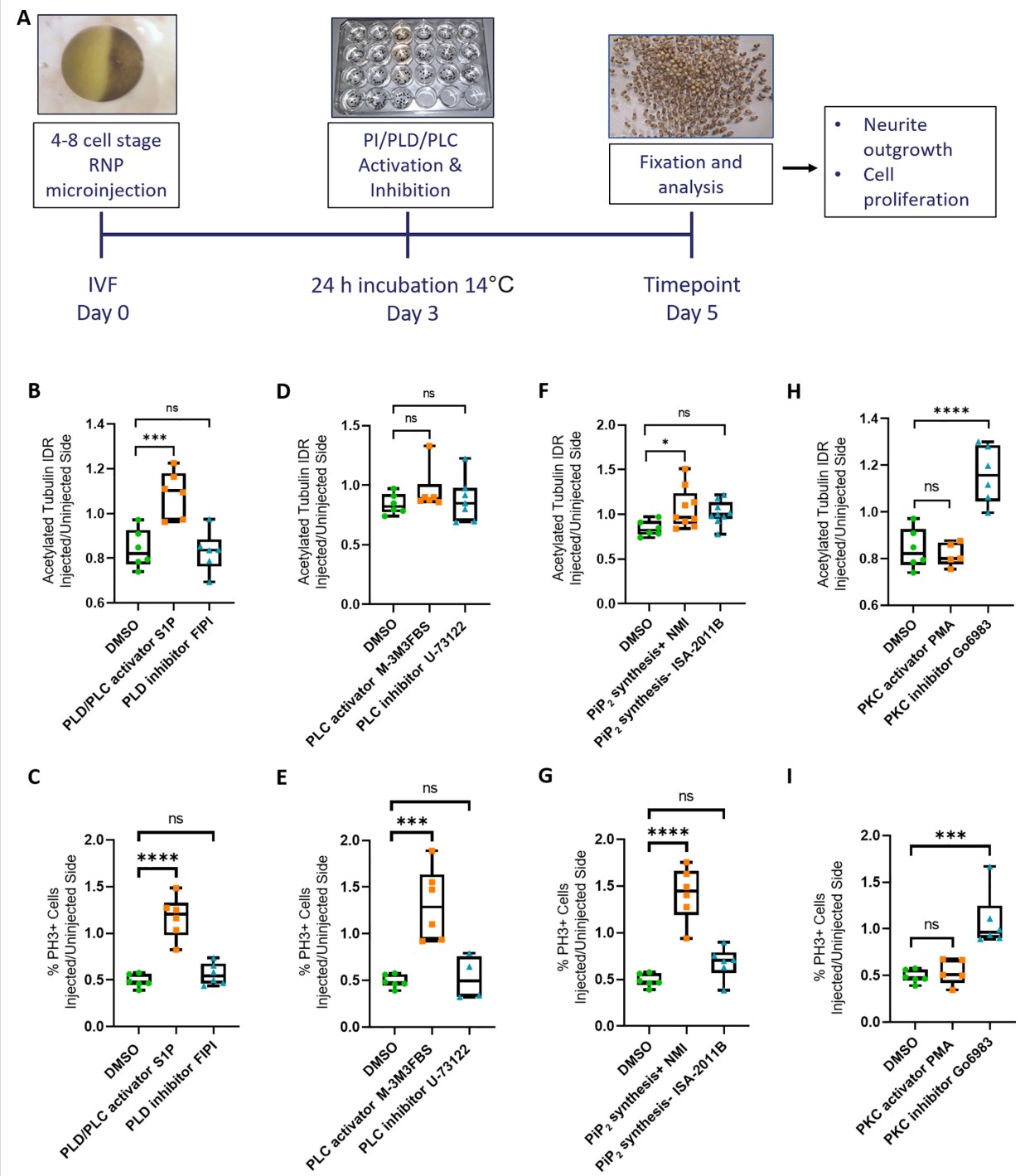

**Figure 4.** Rescue of Marcks and Marcksl1 knockout phenotype by activating phospholipid signaling. (**A**) Experimental flow of the pharmacological study. A dorsal blastomere of 4- to 8-cell stage embryos was co-injected with 500 pg of each single-guide RNA (*marcks.L/S* and *marcksl1.L/S* sgRNA), 1 ng Cas9 protein and 125 pg myc-GFP mRNA to mutate all four homeologs (4M CRISPR). Embryos were then incubated in compounds modulating phospholipase D (PLD)/phospholipase C (PLC) signaling (**B, C**), PLC signaling (**D, E**), $PIP_2$ synthesis (**F, G**), or protein kinase C (PKC) activity (**H, I**) and analyzed for neurite density (integrated density ratio or integrated density ratio (IDR) of acetylated tubulin on injected versus uninjected sides) and mitotic activity (ratio of the proportion of PH3+ cells on injected versus uninjected sides). Significance was determined using an ordinary one-way ANOVA with Dunnett's multiple comparisons test. NS, not significant; *p < 0.05; ***p < 0.001; ****p < 0.0001. Error bars represent minimum to maximum values showing all points with the line at the median. Neurite density and mitotic activity were significantly increased (rescued) in the spinal cord of 4M-CRISPR embryos by activators of phospholipid signaling, enhancers of $PIP_2$ synthesis, and PKC inhibitors. In contrast, inhibitors of phospholipid signaling, inhibitors of $PIP_2$ synthesis, and PKC activators had no effect.

*Figure 4 continued on next page*

*Figure 4 continued*

The online version of this article includes the following figure supplement(s) for figure 4:

**Figure supplement 1.** Effects of modulation of phospholipid signaling on uninjected and 4M CRISPR-injected spinal cord.

cells (*Figure 4H*) on the CRISPR-injected side relative to the uninjected side. In contrast, the PKC activator PMA had no significant effect (*Figure 4G, H*). These findings suggest that the unphosphorylated, presumably membrane-bound fraction of Marcks and Marcksl1 is primarily required for neurite formation and progenitor proliferation in the spinal cord. Our other findings suggest that membrane-bound Marcks and Marcksl1 may promote these processes via the activation of $PIP_2$-mediated PLC and/or PLD signaling. However, the hierarchy of these interactions needs to be confirmed in future studies.

## Expression of Marcks and Marcksl1 during spinal cord regeneration in uninjected tadpoles and in 4M F0-CRISPants

After complete transection of the spinal cord, tadpoles of *X. laevis* have previously been shown to close the injury gap with cells and regenerating neurites at around 1o days post transection (DPT), with almost full functional recovery at 20 DPT (*Lee-Liu et al., 2014*; *Muñoz et al., 2015*; *Edwards-Faret et al., 2018*). Because Marcks and Marcksl1 were identified in a proteomic screen for proteins differentially upregulated after SCI in regeneration-competent tadpoles (*Kshirsagar, 2020*), we next focused on the role of these proteins during spinal cord regeneration in *Xenopus*.

For these experiments, *marcks.L/S* and *marcksl1.L/S* sgRNAs (4M CRISPR) were injected into 1-cell stage embryos, which were then raised to stage 50 tadpoles (*Figure 5A*). After removing some dorsal skin, muscle, and meninges, the spinal cord was transected entirely at a mid-thoracic level. Sham-injured tadpoles were treated the same way but without transection of the spinal cord. Tadpoles were screened before SCI and at 2-, 5-, and 10-DPT for swimming capacity and some tadpoles at each time point were fixed, cryosectioned and evaluated by assessing the distribution of neurites (acetylated tubulin+) and measuring spinal cord gap closure length as well as percentages of proliferating cells (EdU+) and neuro-glial progenitor cells (Sox2+) (*Figure 5A*).

We first analyzed the distribution of Marcks and Marcksl1 in the uninjured spinal cord and at 2-, 5-, and 10-DPT in normal (uninjected) and 4M CRISPR-injected tadpoles. In uninjected, uninjured tadpoles, Marcks protein was detected primarily in the meningeal layer surrounding the spinal cord, with relatively little immunostaining in the spinal cord itself (*Figure 5B–D*). Barely any Marcks protein was detectable in uninjured 4M CRISPR tadpoles (*Figure 5N, O*). At 2 DPT, the levels of Marcks protein in the meninges and spinal cord remain relatively unchanged in both normal (*Figure 5E–G, V*) and CRISPR-injected (*Figure 5P, Q, V*) tadpoles. However, additional Marcks staining was observed in normal tadpoles in cells within the injury gap (*Figure 5E–G*). At 5 DPT, normal tadpoles showed an apparent upregulation of Marcks in all layers of the spinal cord adjacent to the injury site, including the ventricular zone, where neuro-glial progenitors reside, the mantle zone, which harbors differentiated neurons, and the marginal zone, occupied by neurites (*Figure 5H–J*). Marcks also continues to be elevated in cells in the region of the injury gap, which is about to close (*Figure 5H–J*). This pattern of elevated Marcks staining was maintained at 10 DPT (*Figure 5K–M, V*). In contrast, 4M CRISPR tadpoles showed no significant increases of Marcks immunostaining in the spinal cord or adjacent tissues at 5- and 10-DPT (*Figure 5R–U*).

Compared to Marcks, Marcksl1 showed slightly stronger immunostaining in the spinal cord and weaker staining in the meninges of uninjected uninjured tadpoles but less pronounced upregulation after SCI (*Figure 5—figure supplement 1A–L, U*). As described for Marcks, Marcksl1-immunopositive cells also accumulated within the injury gap from 2 DPT onwards (*Figure 5—figure supplement 1D–L*). Again, almost no Marcksl1 immunostaining was detectable in 4M CRISPR tadpoles in and around the spinal cord before the injury, and no significant changes were observed following SCI (*Figure 5—figure supplement 1M–T, U*). Patterns of Marcks and Marcksl1 immunostaining for sham-operated tadpoles resembled those observed in uninjected and uninjured tadpoles and did not change at days 2–10 post-injury (*Figure 5—figure supplement 2*). Taken together, these observations suggest that Marcks and Marcksl1 are present at low levels in the tadpole spinal cord but become upregulated in

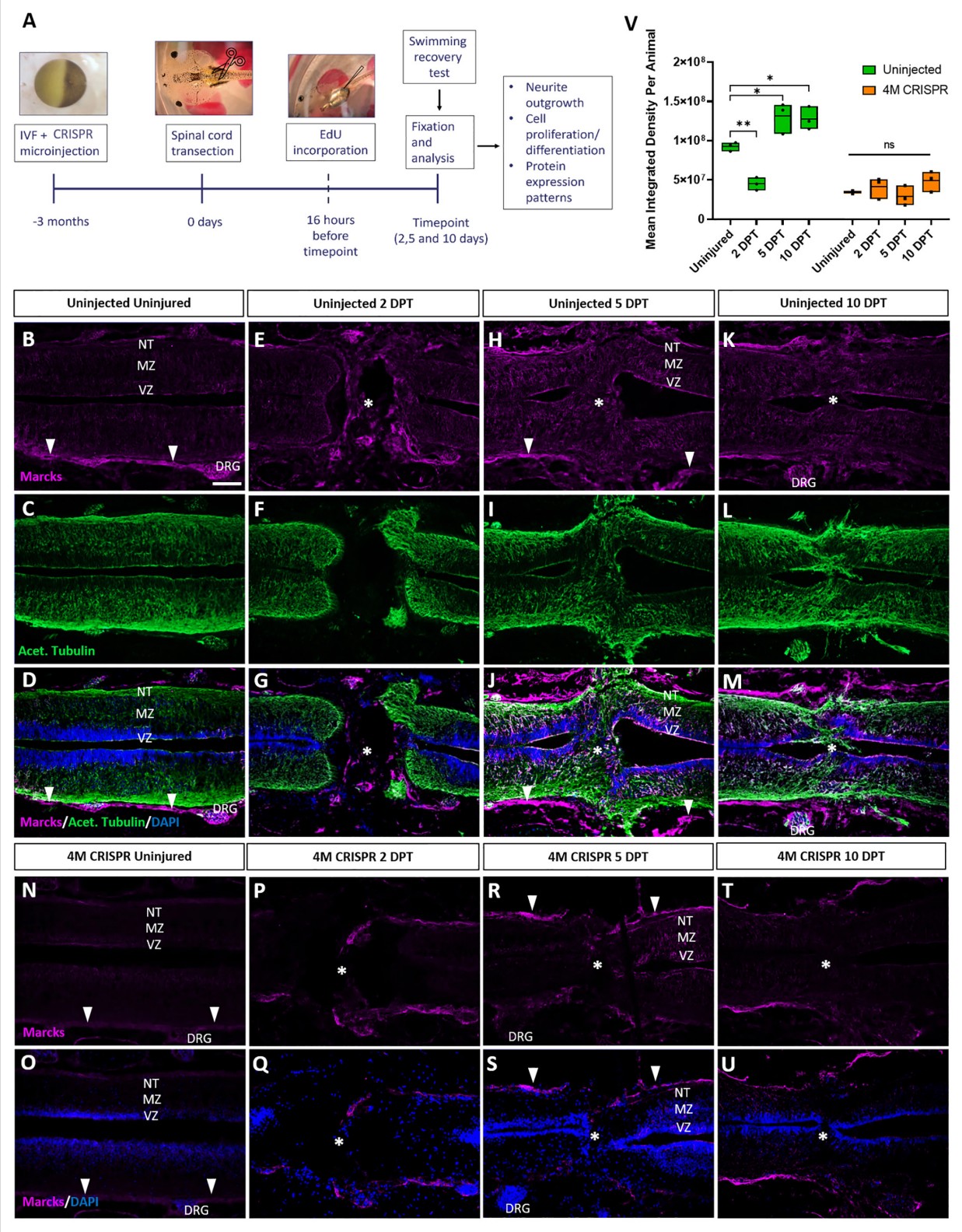

**Figure 5.** Upregulation of Marcks proteins after spinal cord transection is abolished in F0-CRISPants of Marcks and Marcksl1. (**A**) Experimental flow of the spinal cord injury study. One-cell stage embryos were injected with CRISPR constructs targeting all homologs of Marcks and Marcksl1. After approximately 3 months, the spinal cords of uninjected and CRISPR tadpoles (stage 50) were transected. Tadpoles underwent a swimming recovery test at 2-, 5-, and 10-day post transection (DPT). They were analyzed by immunostaining for regenerative outcomes, including neurite outgrowth and

*Figure 5 continued on next page*

*Figure 5 continued*

injury gap closure, cell proliferation (after prior EdU injection), and neuro-glial progenitor markers. Immunostaining for Marcks in horizontal sections of uninjected stage 50 tadpoles without injury (**B–D**) or at 2 DPT (**E–G**), 5 DPT (**H–J**), and 10 DPT (**K–M**). Before the injury, Marcks immunostaining was weakly detected in the ventricular zone (VZ), mantle zone (MZ), and the neurite tracts (NT) of the marginal zone of the spinal cord as well as in the dorsal root ganglia (DRG) and meninges (white arrowheads). After spinal cord transection, it was upregulated in all layers of the spinal cord cells surrounding the injury gap and in cells infiltrating the injury site (white asterisks). Scale bar = 50 μm. *N* = 6 tadpoles per condition. Immunostaining for Marcks in horizontal sections of 4M CRISPR-injected stage 50 tadpoles without injury (**N, O**) or at 2 DPT (**P, Q**), 5 DPT (**R, S**), and 10 DPT (**T, U**). Barely any Marcks immunostaining is detectable at any time point. *N* = 6 tadpoles per condition. (**V**) Quantification of Marcks immunostaining. Significance was determined using an ordinary one-way ANOVA with Tukey's multiple comparisons test. NS, not significant; *$p < 0.05$; **$p ≤ 0.01$. Error bars represent minimum to maximum values showing all points with the line at median. Scale bar in **B** = 50 μm (for all panels).

The online version of this article includes the following figure supplement(s) for figure 5:

**Figure supplement 1.** Marcksl1 proteins are upregulated after spinal cord transection in uninjected but not in 4M CRISPR-injected tadpoles.

**Figure supplement 2.** Marcks and Marcksl1 proteins do not change in uninjected sham-operated tadpoles.

all layers of the spinal cord next to the injury site after SCI (in particular, Marcks), while Marcks- and Marcksl1-immunopositive cells infiltrate the lesion site.

## Marcks and Marcksl1 are required for functional recovery after spinal cord transections in *Xenopus tadpoles*

To further elucidate the role of Marcks and Marcksl1 in spinal cord regeneration, we next used a behavioral test to analyze whether functional recovery after SCI is impaired in 4M CRISPR tadpoles compared to normal (uninjected) tadpoles. Tadpoles were put into 6-well plates equipped with an intermittently vibrating motor to stimulate movement and were video recorded for 1.5 min. Their total swimming distances were then calculated from the videos. The distances covered by uninjected and 4M CRISPR-injected uninjured tadpoles were not significantly different (*Figure 6A*). This, together with our histological data described below, suggests that while the CRISPR editing of Marcks and Marcksl1 perturbs neurite formation and cell proliferation during normal development, ultimately, these tadpoles develop a relatively normally functioning spinal cord. Following SCI, there is a significant decrease in swimming distances at 2 and 5 DPT with no significant difference between uninjected and CRISPR animals (*Figure 6A*, *Figure 6—figure supplement 1A*).

However, at 10 DPT, the average swimming distance of uninjected tadpoles had recovered to a level not significantly different from pre-injury levels. In contrast, the average swimming distance of 10 DPT CRISPR tadpoles was significantly shorter (*Figure 6*, *Figure 6—figure supplement 1*). Animals with sham surgeries showed no significant changes in swimming distances compared to uninjected and uninjured tadpoles over time (*Figure 6—figure supplement 1B*), suggesting that spinal cord transection alone accounts for these behavioral differences. This indicates that the CRISPR editing of Marcks and Marcksl1 significantly delays the capacity of tadpoles to recover functionally after SCI, supporting the vital role of these proteins for spinal cord regeneration.

## Marcks and Marcksl1 are required for injury gap closure and proliferation of neuro-glial progenitors during spinal cord regeneration

We next used immunostaining with acetylated tubulin and DAPI nuclear staining to analyze how *marcks* and *marcksl1* gene editing affects the regrowth of neurites and injury gap closure (measured as the distance between the DAPI+ ventricular zones of rostral and caudal stump) after SCI. Horizontal sections of the spinal cords of uninjured 4M CRISPR-injected animals resembled those of uninjured animals with prominent and well-distinguished ventricular, mantle, and marginal zones (*Figure 6B, F*). This demonstrates that the CRISPR-induced defects during spinal cord development described above only delay but ultimately do not prevent the formation of an adequately organized spinal cord, allowing the use of these animals for spinal cord regeneration studies.

After SCI, the length of the injury gap is initially similar for uninjected and 4M CRISPR-injected tadpoles (*Figure 6C, G, J*). However, at 5 DPT, the first axonal bridges seem to form and connect the rostral and caudal stumps in the uninjected animals, which is delayed until 10 DPT in the CRISPR tadpoles (*Figure 6D, E, H, I*). Moreover, while the injury gap decreased over time in both uninjected and 4M CRISPR-injected tadpoles, the injury gap at 5- and 10-DPT in 4M CRISPR-injected remained

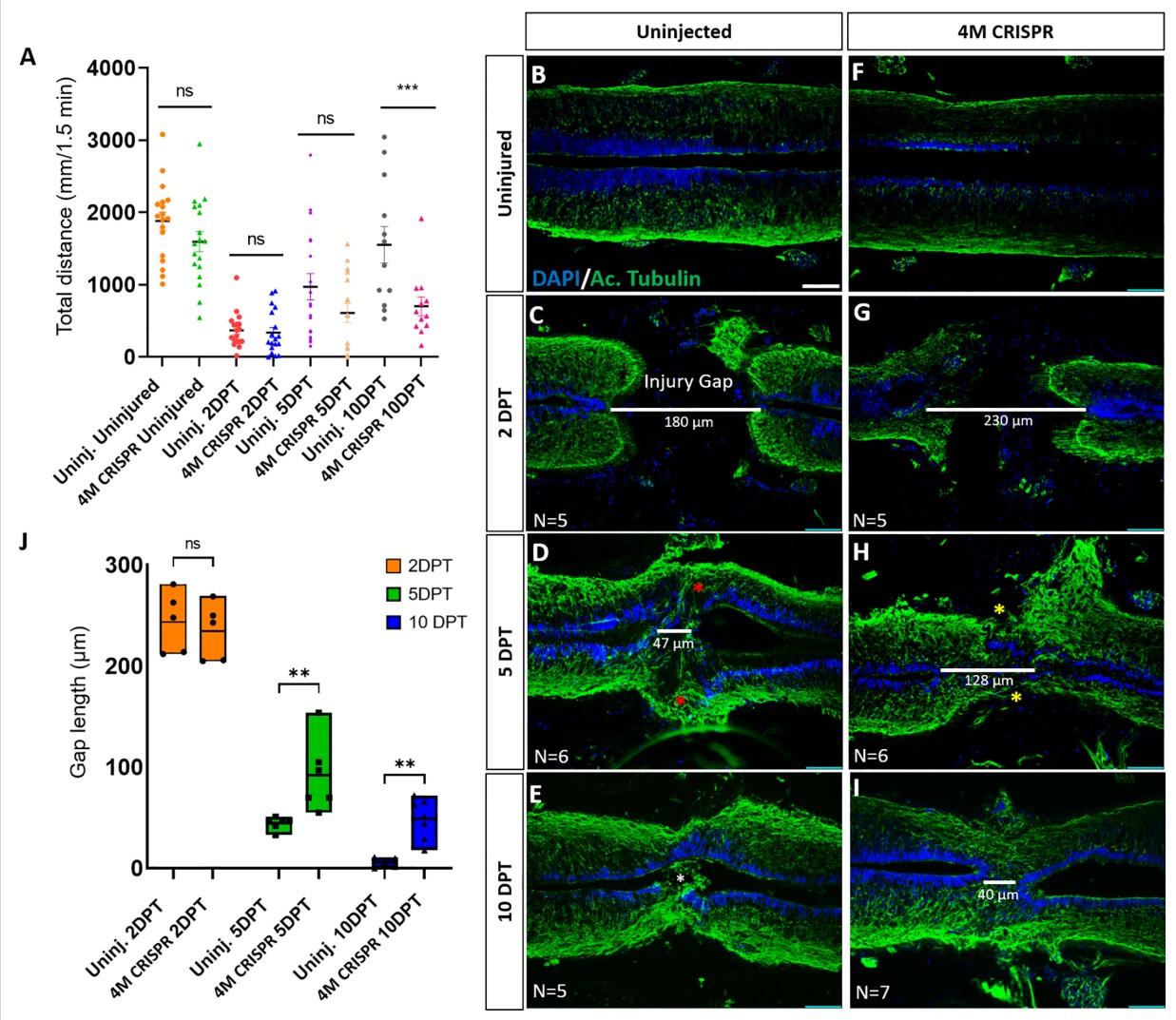

**Figure 6.** Functional recovery and injury gap closure after spinal cord transection are compromised in F0-CRISPants of Marcks and Marcksl1. (**A**) Swimming recovery test in stage 50 tadpoles after spinal cord injury. The graph shows the total swimming distance in uninjected and 4M CRISPR-injected animals before the injury and at 2-, 5-, and 10-DPT. The y-axis indicates the distance traveled by the tadpoles in millimeters for 90 s. Significance was determined using an ordinary one-way ANOVA with an uncorrected Fisher's least significant difference (LSD) test for multiple comparisons. NS, not significant; ***p < 0.001. All data points are shown with the line at mean with SEM. Error bars represent the standard error of the mean. (**B–J**) Comparison of injury gap closure between uninjected and 4M CRISPR-injected stage 50 tadpoles after spinal cord transection. Immunostaining for acetylated tubulin marking axonal tracts with 4',6-diamidino-2-phenylindole (DAPI) marking nuclei in horizontal sections of the spinal cord (**B–I**) and quantification of injury gap length (**J**). Gap length was measured as the distance between the ventricular zones of rostral and caudal spinal cord stumps, as indicated by the white lines. At 2 DPT, an injury gap is visible in uninjected (**C**) and 4M CRISPR-injected (**G**) tadpoles. Axonal bridges connecting the rostral and caudal injury gap in the uninjected tadpoles at 5 DPT (red asterisks) are absent in the 4M CRISPR animals (yellow asterisks). The central canal in uninjected tadpoles becomes continuous at 10 DPT (**E**), in contrast to the 4M CRISPR-injected tadpoles (**I**). Significance was determined using a two-way ANOVA with uncorrected Fisher's LSD of multiple comparisons. NS, not significant; **p ≤ 0.01. All data points are shown with the line at the mean. Scale bar in **B** = 50 μm (for **B–I**).

The online version of this article includes the following figure supplement(s) for figure 6:

**Figure supplement 1.** Functional recovery after spinal cord transection is compromised in F0-CRISPants of Marcks and Marcksl1.

significantly larger than in uninjected embryos, indicating a delay in regenerative closure (*Figure 6D, E, H, I, J*).

To determine the effects of Marcks and Marcksl1 loss of function on cell proliferation during spinal cord regeneration, EdU was injected into the tadpoles 16 hr before the analysis timepoint. DAPI-stained nuclei and EdU+ nuclei were then counted in a defined window across the injury site (i.e., the

area shown in each of the panels of *Figure 7*). There was no significant difference between uninjured uninjected and 4M CRISPR-injected animals (*Figure 7 A1, B1, A2, B, I*). In response to SCI, then a significant increase in the proportion of EdU+ cells was observed in uninjected tadpoles at 2- and 5-DPT, which returned to pre-injury levels at 10 DPT (*Figure 7A1–H, I, Figure 7—figure supplement 1*). An increase in EdU+ cell numbers was seen in the ventricular zone and cells in and adjacent to the injury gap (*Figure 7C1–F1*). Sham-operated animals showed no significant changes in EdU+ cells over time (*Figure 7—figure supplement 2*), indicating that these changes occur only after spinal cord transection. In contrast, there was no significant change in the proportion of EdU+ cells in the 4M CRISPR-injected tadpoles after SCI (*Figure 7A2–H2, Figure 7—figure supplement 1*).

To test whether Marcks and Marcksl1 also affect the distribution and numbers of neuro-glial progenitors during spinal cord regeneration, we then visualized and quantified Sox2-immunopositive cells after SCI. Again, the distribution of Sox2+ cells in the ventricular zone was similar, and there was no significant difference between uninjured uninjected and 4M CRISPR-injected animals (*Figure 8 A1, B1, A2, B2, I*).

In uninjected tadpoles at 2 and 5 DPT, we observed an increase in Sox2+ cell numbers rostral to the injury site of the transected spinal cord (*Figure 8C1–F1*). However, near the injury, Sox2+ cells were reduced (particularly at 2 DPT). This was reflected in a slight decrease in the number of Sox2+ cells counted in our standard window around the injury site (*Figure 8I*). However, at 10 DPT, Sox2+ cells were noticeably and significantly increased in the entire ventricular zone of the regenerating spinal cord, both rostral and caudal of the injury site (*Figure 8, Figure 8—figure supplement 1*). Again, animals with sham surgeries did not show any significant changes in the population of Sox2+ cells (*Figure 8—figure supplement 2*), indicating that the observed changes occur only after spinal cord transection.

In the 4M CRISPR-injected tadpoles, there was a slight overall decrease in Sox2+ cells at 2- and 5-DPT, similar to the uninjected tadpoles (*Figure 8C2–F2, I, Figure 8—figure supplement 1*). However, in contrast to the uninjected tadpoles, the CRISPR tadpoles did not show any elevation of Sox2+ immunostaining in the rostral spinal cord, and there was no overall increase in Sox2+ cell numbers at 10 DPT, with Sox2+ counts significantly decreased compared to uninjected tadpoles (*Figure 8G2, H2, I, Figure 8—figure supplement 1*). Taken together, our findings show that after SCI Marcks and Marcksl1 are required for multiple processes implicated in spinal cord regeneration (*Lee-Liu et al., 2013; Alunni and Bally-Cuif, 2016; Rodemer et al., 2020*), including neurite outgrowth and the proliferation and possibly maintenance of neuro-glial progenitors in the ventricular zone.

## Phospholipid signaling mediates Marcks and Marcksl1 functions during spinal cord regeneration

Since we found phospholipid signaling to be involved in mediating the effects of Marcks and Marcks1 on neurite formation and proliferation of neuro-glial progenitors during normal spinal cord development, we next asked whether the role of Marcks and Marcks1 for spinal cord regeneration may also be modulated by phospholipid signaling. 4M CRISPR-injected tadpoles with SCI were exposed to the activator of phospholipid signaling S1P, the PLD inhibitor FIPI, or only DMSO in control embryos for 7 days. At 7 DPT, the tadpoles were then compared for behavioral recovery, injury gap closure, and EdU+ and Sox2+ cell numbers (*Figure 9*). The swimming recovery test revealed that tadpoles exposed to S1P swam significantly longer distances than control animals (*Figure 9P*). Conversely, tadpoles exposed to the PLD inhibitor FIPI traveled a shorter distance than the DMSO controls, but this difference was insignificant (*Figure 9P*). Immunostaining for acetylated tubulin combined with nuclear DAPI staining showed that injury gap length was significantly reduced in animals incubated in S1P. In contrast, it was significantly higher in the animals incubated in the PLD inhibitor FIPI (*Figure 9A–C, Q*). The proportion of proliferative EdU+ cells (*Figure 9D–I, R*) and of Sox2+ neuro-glial progenitor cells (*Figure 9J–O, S*) was also significantly increased in the tadpoles incubated in S1P and significantly decreased in those exposed to the PLD inhibitor FIPI compared to DMSO controls. Thus, in all parameters investigated, 4M CRISPR tadpoles treated with an activator of phospholipid signaling adopted a phenotype resembling normal tadpoles after SCI. In contrast, those treated with a PLD inhibitor showed more severe loss of function phenotypes than untreated CRISPR tadpoles. In distinction to embryos, where FIPI has no effect, the latter finding suggests that residual PLD activity must be present in CRISPR tadpoles either because some Marcks/Marcksl1-independent

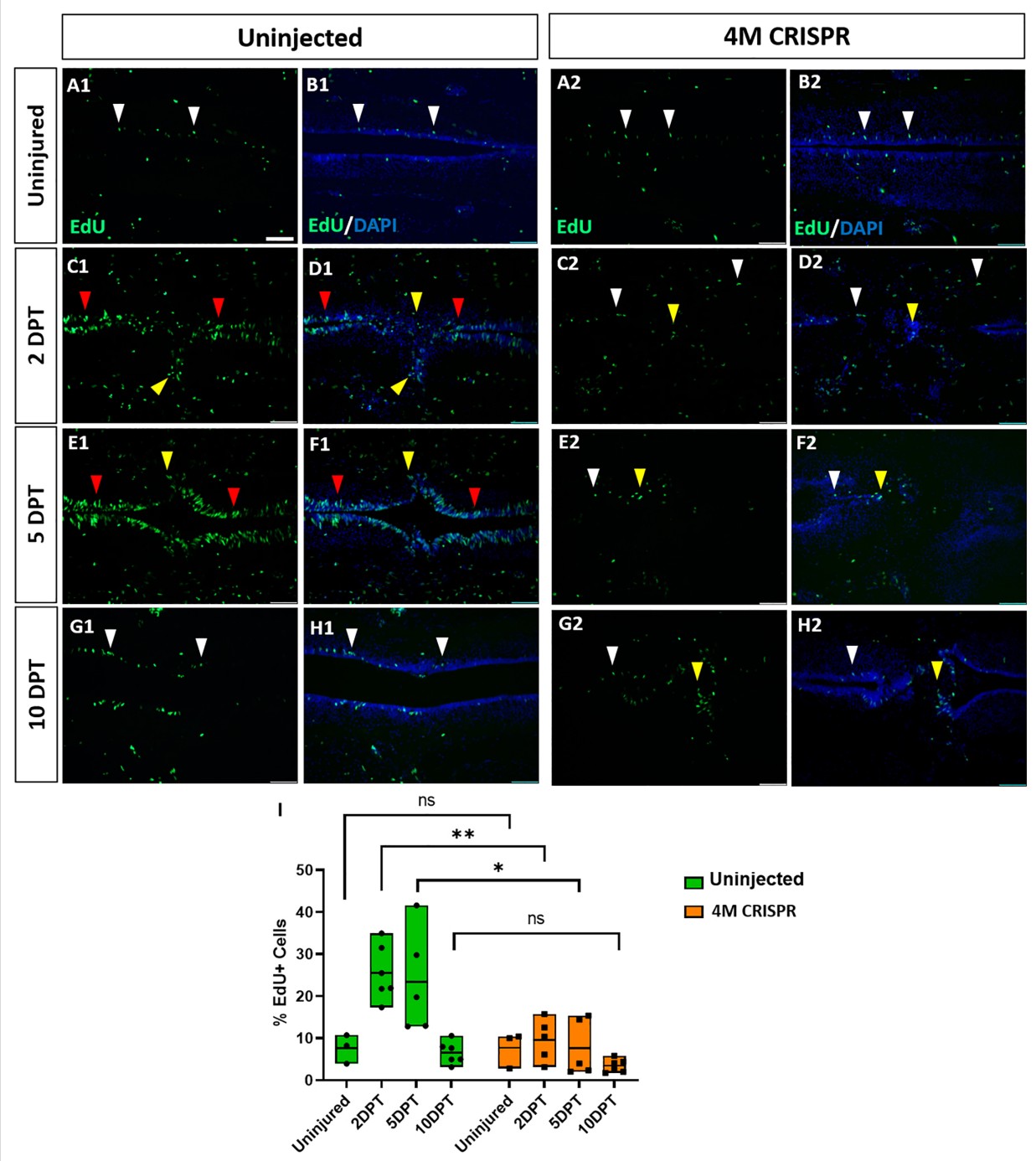

**Figure 7.** Upregulation of cell proliferation after spinal cord transection is compromised in F0-CRISPants of Marcks and Marcksl1. (**A–H**) Evaluation of EdU+ proliferative cells at 2-, 5-, and 10-DPT in horizontal sections of uninjected (**A1–H1**) and 4M CRISPR-injected tadpoles (**A2–H2**) at stage 50 (**A–H**) and quantification of the average proportion of EdU+ cells in a defined window of the spinal cord adjacent to the injury site (**I**). White and red arrowheads indicate preinjury and elevated levels of EdU+ cells in the ventricular zone, respectively. Yellow arrowheads indicate EdU+ cells in and adjacent to the injury gap. Note the increase in EdU+ cells at 2 DPT and 5 DPT in uninjected (**C1–F1, I**) but not 4M CRISPR-injected (**C2–F2, I**) tadpoles. Significance was determined using a two-way ANOVA with Sidak's multiple comparisons test. NS, not significant; *p < 0.05; **p ≤ 0.01. All data points are shown with the line at mean. Scale bar in **A1** = 50 μm (for all panels).

The online version of this article includes the following figure supplement(s) for figure 7:

**Figure supplement 1.** Upregulation of cell proliferation after spinal cord transection is compromised in F0-CRISPants of Marcks and Marcksl1.

**Figure supplement 2.** Sham-operated animals show no alterations of proliferation.

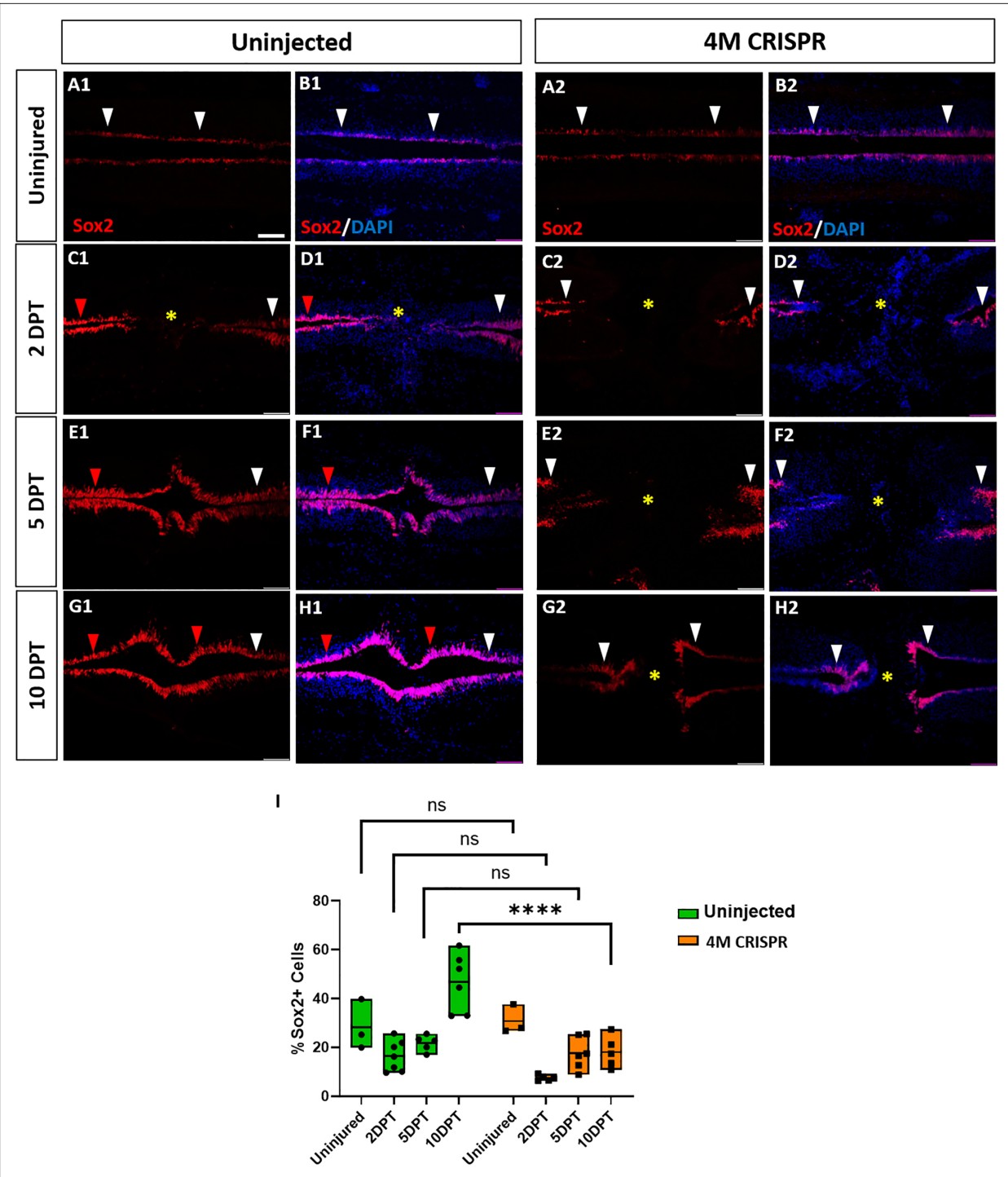

**Figure 8.** Expansion of Sox2+ neuro-glial progenitors after spinal cord transection is compromised in F0-CRISPants of Marcks and Marcksl1. (**A–H**) Evaluation of Sox2+ neuro-glial progenitors at 2-, 5-, and 10-DPT in horizontal sections of uninjected (**A1–H1**) and 4M CRISPR-injected tadpoles (**A2–H2**) at stage 50 (**A–H**) and quantification of the average proportion of Sox2+ cells in a defined window of the spinal cord adjacent to the injury site (**I**). White and red arrowheads indicate preinjury and elevated levels of Sox2+ cells in the ventricular zone, respectively. A yellow asterisk indicates the absence of Sox2+ cells in and adjacent to the injury gap. Note the increase in Sox2+ cells in the spinal cord rostral to the injury gap at 2 DPT (**C1, D1**) and the increase in the proportion of Sox2+ cells at 10 DPT (**G1, H1, I**) in uninjected but not 4M CRISPR-injected tadpoles (**C2, D2, G2, H2, I**). Significance was determined using a two-way ANOVA with Sidak's multiple comparisons test. NS, not significant; ****$p < 0.0001$. All data points are shown with the line at mean. Scale bar in **A1** = 50 μm (for all panels).

The online version of this article includes the following figure supplement(s) for figure 8:

*Figure 8 continued on next page*

**Figure supplement 1.** Upregulation of Sox2+ neuro-glial progenitors after spinal cord transection is compromised in F0-CRISPants of Marcks and Marcksl1.

**Figure supplement 2.** Sham-operated animals show no alterations of numbers of neuro-glial progenitors.

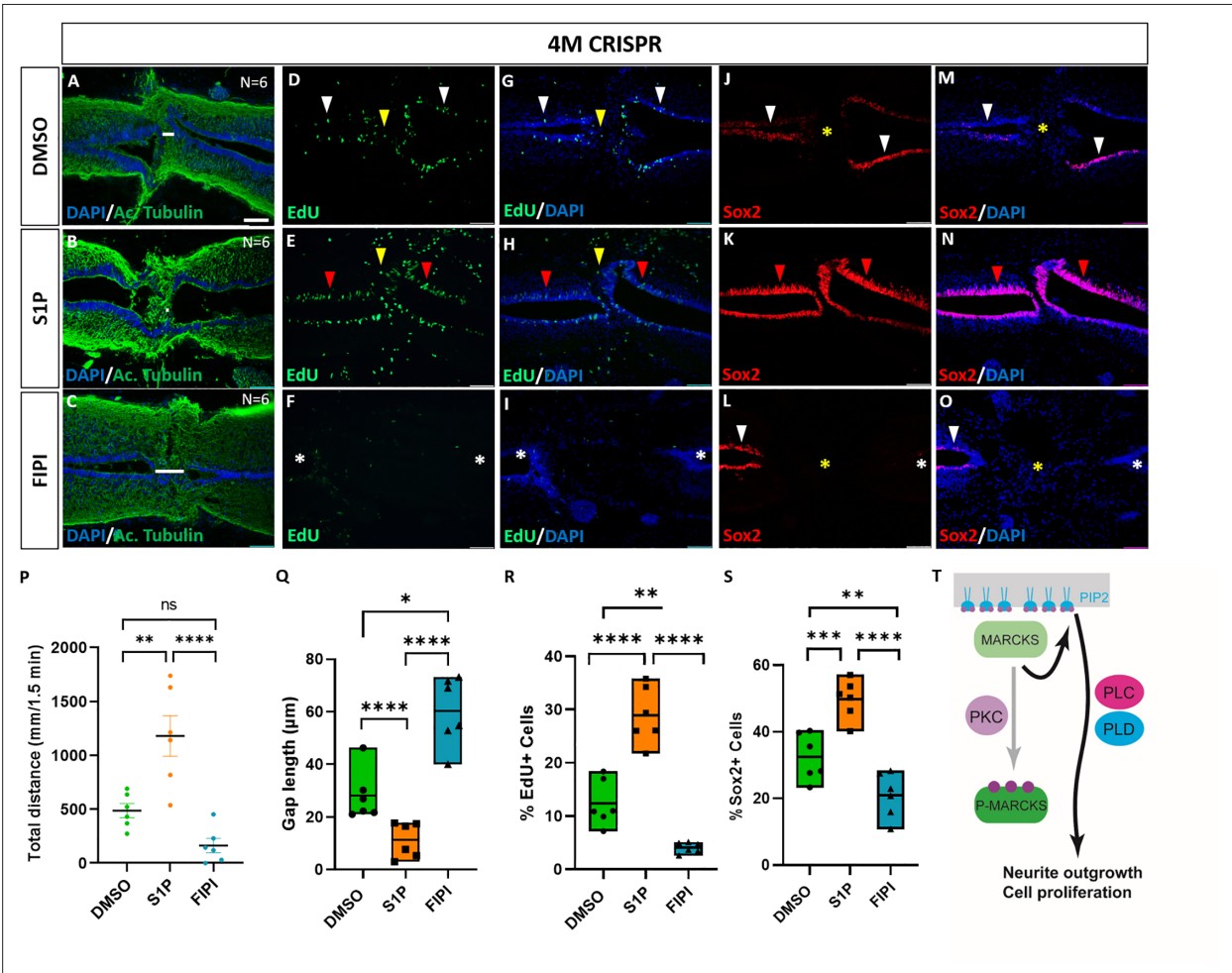

**Figure 9.** Activation of phospholipid signaling rescues deficiencies of spinal cord regeneration in 4M CRISPR-injected tadpoles. (**A–O**) Horizontal sections of 4M CRISPR-injected tadpoles at stage 50 exposed to DMSO only (controls; **A, D, G, J, M**) or to PLD/PLC activator Sphingosine-1-phosphate (S1P) (**B, E, H, K, N**) or PLD inhibitor FIPI (**C, F, I, L, O**) for 7 DPT and analyzed for injury gap closure (**A–C**); the proportion of proliferative EdU+ cells (**D–I**) and the proportion of Sox2+ neuro-glial progenitors (**J–O**). Immunostaining for acetylated tubulin and nuclear DAPI staining was used to visualize injury gap closure (**A–C**). Gap length was measured as the distance between the ventricular zones of rostral and caudal spinal cord stumps, as indicated by the white lines. White and red arrowheads indicate preinjury and elevated levels of EdU+ or Sox2+ cells in the ventricular zone, respectively. Yellow arrowheads indicate EdU+ cells in and adjacent to the injury gap. White asterisks indicate reduced levels of EdU+ or Sox2+ cells in the ventricular zone. A yellow asterisk indicates the absence of Sox2+ cells in and adjacent to the injury gap. Quantification of effects of pharmacological treatments in 4M CRISPR-injected tadpoles at 7 DPT on swimming recovery (**P**), injury gap closure (**Q**), proportion of proliferative EdU+ cells (**R**), and proportion of Sox2+ neuro-glial progenitors (**S**). Note that the activator of phospholipid (PLD/PLC) signaling S1P significantly enhances swimming recovery, injury gap closure, and proportions of proliferating cells (EdU+) and neuro-glial progenitors (Sox2+) after spinal cord transection in 4M CRISPR tadpoles compared to DMSO-treated controls. At the same time, the PLD inhibitor FIPI significantly reduces injury gap closure and proportions of proliferating cells (EdU+) and neuro-glial progenitors (Sox2+). Significance was determined using an ordinary one-way ANOVA with uncorrected Fisher's least significant difference (LSD) for multiple comparisons. $N = 6$ animals per condition. NS, not significant; *$p < 0.05$; **$p \leq 0.01$; ***$p < 0.001$; ****$p < 0.0001$. All data points are shown with the line at mean. Error bars in (**P**) epresent the standard error of the mean. Scale bar in **A** = 50 µm (for all panels). (**T**) Model proposing that unphosphorylated, membrane-bound Marcks and Marcksl1 proteins promote neurite outgrowth and proliferation during spinal cord development and regeneration in a phospholipid-dependent manner.

PLD activity remains or because Marcks/Marcksl1 have not completely been knocked out in the entire spinal cord.

Taken together our findings show that activation of phospholipid signaling can rescue the delaying effects of Marcks and Marcksl1 loss of function on functional recovery, injury gap closure, and proliferation of neuro-glial progenitors after SCI, while PLD inhibition further exacerbates them. This suggests that the functions of Marcks and Marcksl1 during spinal cord regeneration are at least partially mediated by phospholipid signaling, similar to their roles in normal spinal cord development.

## Discussion

The upregulation of Marcks and Marcksl1 proteins after SCI in the regenerating spinal cord of *Xenopus tadpoles*, but their downregulation in the non-regenerative spinal cord of froglets (*Kshirsagar, 2020*) led us to investigate the role of these proteins during normal spinal cord development as well as during spinal cord regeneration. Our findings indicate that these proteins act redundantly and are required for normal neurite formation, proliferation, and possibly the maintenance of neuro-glial progenitor cells during spinal cord development. However, a reasonably normal spinal cord will eventually develop after the knockdown or gene editing of both proteins, suggesting that Marcks- and Marcksl1-independent mechanisms can subsequently compensate for the loss of these proteins.

We also demonstrated that Marcks and Marcksl1 play similar roles in spinal cord regeneration in tadpoles after injury. Neurite regrowth, injury gap closure, activation of proliferation, and increase in the number of Sox2+ neuro-glial progenitors near the injury site were drastically reduced in tadpoles after the gene editing of both proteins and functional recovery of swimming was delayed. The upregulation of proliferating Sox2+ cells in the ventricular zone was previously shown to be required for regeneration, suggesting that increased division and migration of these cells contribute to injury gap closure and the generation of a permissive environment for neurite regrowth (*Gaete et al., 2012*; *Muñoz et al., 2015*). Our findings that Marcks and Marcksl1 are required for proper regeneration of the spinal cord after injury suggest that the downregulation of both proteins in postmetamorphic froglets (*Kshirsagar, 2020*) may be one of the factors preventing spinal cord regeneration in adults. To test this hypothesis, future studies should determine, whether spinal cord regeneration in froglets can be experimentally promoted by overexpression of these proteins.

Our observation that Marcks and, to a lesser extent, Macksl1 are upregulated in the entire spinal cord, including the ventricular zone, as well as in cells filling the injury gap, suggests that these proteins may act cell-autonomously in the spinal cord-derived cells to promote these regenerative processes. This is also supported by a recent single-cell RNA-seq study of regenerating *Xenopus* tails, which showed that Marcks and Marcksl1 are highly enriched in many neural cell types during tail regeneration (*Aztekin et al., 2019*). However, since both proteins are known to be expressed in macrophages and other immune cells and can modulate the inflammatory response (*Carballo et al., 1999*; *El Amri et al., 2018*; *Zhou and Li, 2000*), Marcks- and Marcksl1-immunopositive immune cells that infiltrate the injury site may also contribute to the regeneration-promoting effects of these proteins. Double-labeling with cell type-specific markers and single-cell RNA-seq following SCI will be required in the future to clarify the relative contribution of different Marcks- and Marcksl1-expressing cell types to spinal cord regeneration. Moreover, transplantations of spinal cords between F0-CRISPants and wild-type embryos will allow to address, whether these proteins are required autonomously or non-autonomously for spinal cord regeneration.

We also show here that activation of phospholipid signaling can rescue deficiencies of neurite formation and neuro-glial progenitor proliferation after the loss of function of Marcks and Marcksl1 during spinal cord development and regeneration. In contrast, a specific inhibitor of PLD-dependent phospholipid signaling further enhances these deficiencies after SCI. Taken together with our findings that enhancement of $PIP_2$ synthesis and PLC signaling as well as inhibition of PKC is also able to partially rescue the effects of Marcks and Marcksl1 loss of function during spinal cord development, this is compatible with a model in which unphosphorylated, membrane-bound Marcks promotes neurite outgrowth and progenitor proliferation via $PIP_2$-dependent activation of phospholipid signaling (*Figure 9T*). This is in agreement with previous studies showing that unphosphorylated membrane-bound Marcks promotes neurite outgrowth by modulating $PIP_2$ signaling due to its ability to sequester $PIP_2$ in membrane rafts (*Laux et al., 2000*), thereby potentially stimulating $PIP_2$ signaling (*Trovò et al., 2013*; but see *Laux et al., 2000*). It also provides a link to previous studies suggesting an essential role

of PLD-dependent mTORC1 signaling for the proliferation of neuro-glial progenitors during spinal cord regeneration (*Sun and Chen, 2008*; *Kanno et al., 2012*; *Peñailillo et al., 2021*). However, most pharmacological compounds are known to modulate additional signaling pathways to some extent and the predicted role of phosphorylation and subcellular localization of Marcks/Marcksl1 needs to be validated. Therefore, additional experiments are needed to corroborate the various specific interactions implied by this model.

In conclusion, our present study reveals a hitherto unrecognized essential function of Marcks and Marcksl1 for spinal cord development and regeneration, highlighting the potential of *Xenopus* as a model system for regeneration studies and suggesting new pathways to be targeted for therapeutic stimulation of spinal cord regeneration in human patients.

# Materials and methods

## Animals

Frogs (*X. laevis*) were obtained from a commercial breeder (*Xenopus* One Corp, Michigan, USA). All experiments were done in accordance with Irish and European legislation under licenses AE19125/P079 and AE19125/P098 from the Health Products Regulatory Authority (HPRA).

## Morpholino antisense oligonucleotides

Translation-blocking MOs, and a standard control MO (5'-CCTCTTACCTCAGTTACAA TTTATA-3') were obtained from Gene Tools. *X. laevis* is an allotetraploid (*Session et al., 2016*) and its genome contains two homeologs (denoted .L and .S, respectively) of many genes, including *marcks* and *marcksl1*. Therefore, in our functional studies, we designed MOs to block both homeologs. The marcks MO (5'-GGTCTTGGAGAATTGGGCTCCCA TT-3') (*Iioka et al., 2004*) target base pairs –16 to +9 of both *marcks.S* and *marcks.L*. The marcksl1.L MO (5'-GGATTCTATACTACCCATTTTCCGC-3') targets base pairs –7 to +18 of *marcksl1.L*, while the marcksl1.S MO (5'- GGACTCTACGCTACCCATTGTGACT-3') (*Zhao et al., 2001*) targets base pairs –2 to +23 of *marcksl1.S*.

The efficacy of all MOs was verified in western blots following in vitro transcription and translation (TNT-coupled Wheat Germ Extract kit, Promega) of pCMV-Sport6-marcks.S, pCMV-Sport6-marcksl1.S, and pCMV-Sport6.ccdb-marcksl1.L (1 µg/50 µl reaction) with and without MO (20 pg marcksl1.L MO or 40 pg of marcksL1.S or marcks.S MOs/50 µl reaction) as previously described (*Ahrens and Schlosser, 2005*). 2 µl biotinylated Transcend tRNA (Promega) was included in the reaction mix resulting in biotinylation of lysine residues. Polyvinylidene difluoride (PVDF) membranes were subsequently incubated with Streptavidin Alkaline Phosphatase (1:5000; Promega) to visualize proteins with Transcend Chemiluminescence Substrate (Promega). The specificity of all MOs was verified by co-injection with the rescue mRNAs (see below).

## Expression constructs

Plasmids encoding membrane-GFP (pCS2+-memGFP) and c-myc-GFP (pCMTEGFP) were provided by John Wallingford (*Wallingford et al., 2000*) and Doris Wedlich, respectively. cDNA clones for *X. laevis* marcks.S (clone accession: BC041207), marcks.L (clone accession: BC084888), and marcksl1.S (clone accession: BC157454) were purchased from Source Bioscience, provided in pCMV-Sport6 backbone vectors and pCMV-Sport6.ccdb for *marcksl1.L*. Due to a nucleotide sequence identity of 94% between *marcks.L* and *marcks.S*, only the cDNA clone for *marcks.S* was used. All plasmids were verified by sequencing and linearized with ClaI before mRNA synthesis using the SP6 Ambion Message Machine Kit (Invitrogen).

For rescue experiments, chicken *marcks* (clone ID: OGa33947C) and *Xenopus tropicalis marcksl1* (clone ID: OXa00945C) ORF clones were purchased from GenScript (GenBank accession numbers: NM_001044437.1; XM_015284506.2). The clones were received in pET-23a(+) vectors, containing a His-tag and T7-tag, flanking the cloning sites on each terminal. There were 3/25 base-pair mismatches between Xl-Marcks.S MO and the pET-23a(+)-marcks_OGa33947C sequence; 9/25 mismatches between Xl-Marcksl1.L MO and pET-23a(+)-marcksl1_OXa00945C; and 10/25 mismatches between Xl-Marcksl1.S MO and pET-23a(+)-marcksl1_OXa00945C. To increase the number of mismatches between the MOs and the mRNA rescue constructs, seven and eight additional substitutions were introduced into pET-23a(+)-marcks_OGa33947C and pET-23a(+)-marcksl1_OXa00945C_plasmids,

respectively, using the Q5 site-directed mutagenesis kit (New England Biolabs). These substitutions did not alter protein sequences. For mutagenesis, forward (5′-TTTAGTAAAACAGCTGCGAAGGGCGAAGCC-3′) and reverse (5′- CTGTGCACCCATGGATCCGCGACCCATTTGC-3′) primers for pET-23a(+)-marcks_OGa33947C were annealed at 70°C, while forward (5′- ATCGAGAGTAAGTCTAAGAGTGCAGATATTAGC-3′) and reverse (5′-GGAGCCCATGGATCCGCGACCCATTTG-3′) primers for pET-23a(+)-marcksl1_OXa00945C were annealed at 63°C. Mutagenized plasmids were verified by sequencing and linearized with AdeI before mRNA synthesis using the T7 Ambion Message Machine Kit (Invitrogen), followed by polyadenylation using the Poly(A) Tailing Kit (Invitrogen).

## CRISPR/Cas9-based gene editing

*marcks.L/S* and *marcksl1.L/S* genomic sequences were derived from the most recent version of the *X. laevis* genome on Xenbase (https://www.xenbase.org/) and NCBI (https://www.ncbi.nlm.nih.gov/genome/). Two single-guide RNA primers targeting the first exon of *marcks.L/S* (5′-CTAGCTAATACGACTCACTATAGGCTGCGGTCTTGGAGAATTGTTTTAGAGCTAGAAATAGCAAG-3′) and second exon of *marcksl1.L/S* (5′-CTAGCTAATACGACTCACTATAGGTTGGGGTCCCCGTTGGCGGTTTTAGAGCTAGAAATAGCAAG-3′), respectively (target-specific region underlined), were designed using CRISPRscan (https://www.crisprscan.org). These gene-specific sgRNA primers contained a T7 promoter site (TAATACGACTCACTATA) with five additional nucleotides in the 5′ end for enhanced transcriptional output, two guanines for in vitro transcript yield efficiency, 17–18 nucleotides encoding the target-specific region (underlined), and a 23-nucleotide tail for annealing to the 80-nucleotide universal reverse primer (5′-AAAAGCACCGACTCGGTGCCACTTTTTCAAGTTGATAACGGACTAGCCTTATTTTAACTTGCTATTTCTAGCTCTAAAAC-3′). As a control, an sgRNA targeting *solute carrier family 45 member 2* (*slc45a2*) 5′-CTAGCTAATACGACTCACTATAGGTTACATAGGCTGCCTCCAGTTTTAGAGCTAGAAATAGCAAG-3′, encoding a protein that mediates melanin synthesis, was used (**DeLay et al., 2018**). In *X. laevis*, *slc45a2* is present as only one homeolog, *slc45a2.L*. Off-target, and editing efficiency probabilities were assessed using InDelphi (https://indelphi.giffordlab.mit.edu/), GGGenome (https://gggenome.dbcls.jp/), and CRISPRscan.

sgRNAs were synthesized (**Bhattacharya et al., 2015**) using PCR with Q5 High-Fidelity DNA Polymerase (NEB), 1 µM each of forward and reverse primer and the following cycling conditions: 98°C for 30 s; 10 cycles of 98°C for 10 s, 62°C for 20 s, 72°C for 20 s; 25 cycles of 98°C for 10 s, 72°C for 30 s; and 72°C for 5 min. PCR products were purified using the DNA Clean & Concentrator-5 (Zymo Research, D4014) and in vitro transcribed using the MEGAshortscript T7 Transcription Kit (Invitrogen). After a 4-hr incubation at 37°C and subsequent TURBO DNase treatment for 15 min, the sgRNA was purified using the GeneJET RNA Cleanup and Concentration Micro Kit (Thermo Scientific) in accordance with the manufacturer's protocol with the following modifications: The transcription reaction mixture was first adjusted to 50 µl with nuclease-free water, followed by the addition of 250 µl of Binding Buffer. After elution in nuclease-free water, the concentration of sgRNAs was adjusted to 500 ng/µl, and aliquots were stored at −80°C.

## Microinjections

Embryos of *X. laevis* were staged according to **Nieuwkoop and Faber, 1967** and injected according to standard procedures (**Sive et al., 2000**). mRNAs or MOs were injected into single blastomeres at the 4- to 8-cell stage, giving rise to the dorsal ectoderm. To visualize membranes in confocal microscopy, *mem-GFP* mRNA was injected (125 pg). For the overexpression study, 100 or 300 pg of *marcks* or *marcksl1* mRNA was co-injected with 125 pg *myc-GFP* mRNA to identify the injected side. As a control, only *myc-GFP* mRNA (125 pg) was injected. MOs (see above) were injected singly or as a cocktail (9 ng each) with 125 pg c-*myc*-GFP mRNA. 27 ng of the standard control MO was used in control injections. For MO rescue experiments, 300 pg mutated chicken *marcks* mRNA and 350 pg mutated *Xenopus tropicalis marcksl1* mRNA was co-injected with 9 ng of each MO. For CRISPR/Cas9 experiments, embryos were injected into single blastomeres at the 1- to 8-cell stage (targeted to the dorsal blastomere in injections at the 4- to 8-cell stage) with a ribonucleoprotein (RNP) mix containing 500 pg of each sgRNA, 1 ng TrueCut Cas9 Protein v2 (Invitrogen, A36497), and 125 pg *myc-GFP* mRNA. The RNP mix was incubated at room temperature for at least 10 min before injection.

## Extraction and analysis of genomic DNA

Approximately 5 days after the injection of RNPs, genomic DNA was extracted from embryos using the Wizard Genomic DNA Purification Kit (Promega) according to the manufacturer's Animal Tissue protocol. Upon extraction, the DNA was rehydrated in 25 µl of DNA Rehydration Solution overnight at 4°C. CRISPR-targeted regions in *marcks.L/S* and *marcksl1.L/S* were PCR-amplified using the EmeraldAmp GT PCR Master Mix (Takara Bio). For every gene, each 40 µl reaction contained 20 µl of 2× EmeraldAmp Master Mix, 1 µl of the forward and reverse primer mix diluted to 10 µM, 100 ng of genomic template DNA, and PCR-grade $H_2O$ for volume adjustment. The cycling conditions were 98°C for 30 s, 30 cycles of 98°C for 10 s, 60°C for 30 s, 72°C for 1 min per kilobase of PCR product, and 72°C for 5 min. The following forward (F) and reverse (R) primer pairs were used: *marcks.L* (F: 5'-ATCACCTGATGGACGCATGG-3', R: 5'-CCCCCACATCTAAAGCGGAG-3'), *marcks.S* (F: 5'-AGTGTCATGAATCAGCGGGG-3', R: 5'-GTGTGTCTATTAGCGGCGGA-3'), *marcksl1.L* (F: 5'-GCTAGGAAGAAGCGAGTCCC-3', R: 5'-ACCCGTTAACATGAGCAGCA-3'), *marcksl1.S* (F: 5'-AGAAAGAAGTGAGCCTAGAGTGATT-3', R: 5'-ATCCCTCCAAGGGTGACAGG-3'), *slc45a2.L* (F: 5'-GTTCCCTTCGCTCATACAATG-3', R: 5'-GCCAGAAAGGGGTTTATTGC-3'). 5 µl of each PCR product was run in a 1% agarose gel and the remaining 35 µl were purified using the QIAquick PCR Purification Kit (QIAGEN) before being sent for sequencing (sequencing primers: *marcks.L*: 5'-CCCATGCTGTCTGTCTTTGA-3', *marcks.S*: 5'-GTGTGTCTATTAGCGGCGGA-3', *marcksl1.L*: 5'-TCTTCTTCTGGCGCCTGC-3', *marcksl1.S*: 5'-AGTTTAGGGAGGCAGGGTTG-3', *slc45a2.L*: 5'-GTTCCCTTCGCTCATACAATG-3'). Sequencing raw data from wild-type, *slc45a2*, and *marcks/marcksl1*-targeted embryos were analyzed for the frequency of insertion/deletion mutations using the Synthego ICE algorithm (https://ice.synthego.com/) (**Conant et al., 2022**).

## Antibodies

Affinity-purified polyclonal antibodies against *Xenopus* Marcks and Marcksl1 proteins were generated by the company GenScript. For anti-Marcks, a His-tagged fusion protein with the following sequence was expressed in bacteria: MHHHHHHASPAEGEPAEPASPAEGEPAAKTEEAGSTSTPSTSNETPKKKKKR FSFKKSFKLSGFSFKKNKKENSEGAENEGA. Using the Protein BLAST NCBI tool, this immunogen (including the 6-His-tag) shared a 98.68% identity (*E* value: 3e−43) with Marcks.L (NCBI Reference Sequence XP_018118779.1) and 98.67% (*E* value: 3e−42) with Marcks.S (XP_018119725.1). No other sequences in *X. laevis* (taxonomy id:8355) returned any significant alignments. The protein was purified in Nickel and Superdex 75 columns and, after liquid chromatography–mass spectrometry validation, was used to immunize rabbits. The antibody was then antigen-affinity-purified from sera and tested by indirect ELISA by Genscript. For anti-Marcksl1, keyhole limpet hemocyanin (KLH)-tagged peptides (CGDPKPEDPPGKQAK) were used to immunize rabbits and were affinity-purified. This peptide had a 100% sequence identity (*E* value: 6e−08) with both Marcksl1.L (NP_001108274.1) and Marcksl1.S

**Table 1.** Primary antibodies.

| Antigen | Company, product code | Reactivity | Dilution |
|---|---|---|---|
| Acetylated tubulin | Sigma, T6793 | Mouse Monoclonal | 1:1000 |
| Phospho-Histone H3 (PH3) | Merck, 06-570 | Rabbit Polyclonal | 1:200 |
| GFP | Abcam, ab290 | Rabbit Polyclonal | 1:250 |
| GFP | Abcam, ab1218 | Mouse Monoclonal | 1:250 |
| c-Myc | Developmental Studies Hybridoma Bank (DSHB), 9E 10 | Mouse Monoclonal | 1:200 |
| c-Myc | Merck, PLA0001 | Rabbit Polyclonal | 1:200 |
| 6-His-Tag | Abcam, ab3554 | Chicken Polyclonal, FITC | 1:100 |
| T7-Tag | Invitrogen, PA1-32390 | Goat Polyclonal, FITC | 1:200 |
| *X. laevis* Marcks | Genscript, custom-made | Rabbit Polyclonal | 1:200 |
| *X. laevis* Marcksl1 | Genscript, custom-made | Rabbit Polyclonal | 1:100 |
| Sox2 | Cell Signaling Technology (CST), D6D9 | Rabbit Monoclonal | 1:100 |
| Sox3 | *Zhang et al., 2004* | Rabbit Polyclonal | 1:100 |

(NP_001081982.2). The next closest sequence alignments returned insignificant $E$ values greater than 0.34.

## Histochemistry

Embryos and tadpoles were fixed in 4% paraformaldehyde (PFA) in 0.1 M phosphate buffer (PB) overnight, followed by two washes in PB for 10 min, before cryoprotection in 20% sucrose/PB overnight at 4°C. The specimens were then positioned in rubber forms and embedded in O.C.T. (VWR) for 30 min before shock-freezing in 2-methylbutane cooled in liquid nitrogen. Immunohistochemistry on 20 µm cryosections was performed as previously described (*Schlosser and Ahrens, 2004*). Primary antibodies were diluted, as indicated in *Table 1*. The following secondary antibodies were used at 1:500 dilution supplemented with DAPI (100 ng/µl): Alexa Fluor 488- and Alexa Fluor 594-conjugated goat anti-mouse (Invitrogen A11001 and A11005, respectively); Alexa Fluor 488- and Alexa Fluor 594-conjugated goat anti-rabbit (Invitrogen A11008 and A11012, respectively); and Alexa Fluor 594-conjugated donkey anti-mouse and anti-rabbit (Invitrogen A21203 or A21207, respectively).

Incorporation and histochemical detection of 5-ethynyl-2'-deoxyuridine (EdU) was used to label proliferating cells before and after SCI. EdU was initially diluted to 5 mg/ml in 0.8× phosphate-buffered saline (PBS), followed by 5 µl intracoelomic injections into tadpoles 16 hr before the desired time point (*Edwards-Faret et al., 2017*). In tadpole cryosections immunostained for Sox2, EdU was detected after secondary antibody incubation using the Click-iT EdU Cell Proliferation Kit for Imaging and Alexa Fluor 488 dye (Invitrogen).

## In situ hybridization

After fixation in 4% PFA, whole-mount in situ hybridization was carried out as previously described (*Schlosser and Ahrens, 2004*). pcMV-Sport6-marcks.S, pCMV-Sport6-marcksl1.S, and pCMV-Sport6.ccdb-marcksl1.L were linearized with EcoRV followed by synthesis of anti-sense RNA probes using the digoxigenin RNA Labeling Kit (Sigma-Aldrich) and SP6 polymerase. Vibratome sections (40 µm) were cut after the whole-mount in situ hybridization.

## Pharmacological assays

Uninjected and RNP-injected embryos (see above) were incubated in various chemical compounds 3 days post-fertilization for 24 hr at 14°C in a dark incubator. All compounds were solubilized in DMSO and diluted in 0.1× Modified Barth's Solution (MBS) (*Sive et al., 2000*). Control embryos were incubated in 2% DMSO/0.1× MBS. The medium was changed once after 12 hr of incubation in a 12-well plate (6 embryos per well). To determine the proper dosage, a dose-titration screening was first conducted for all chemicals using dilutions of 0.01, 0.1, 1, and 10 µM in DMSO. The highest dose that produced comparable mortality rates to the DMSO control embryos was selected for further studies. S1P (1 µM, TOCRIS) and 5-Fluoro-2-indolyl des-chlorohalopemide (FIPI) (10 µM, TOCRIS) were used to activate and inhibit PLD, respectively. NMI (10 µM, TOCRIS) and ISA-2011B (10 µM, MedChemExpress) enhanced or inhibited phosphatidylinositol biosynthesis, respectively. Finally, PMA (0.01 µM, TOCRIS) and Go 6983 (10 µM, TOCRIS) were used to activate or inhibit PKC, respectively.

## Imaging and statistical analysis

Immunostained cryosections and vibratome sections were visualized using an Olympus BX51 fluorescence microscope and recorded with an Olympus DP72 camera. Some representative images were obtained by confocal microscopy (Olympus Fluoview 1000 Confocal Microscope). For each epitope, images were acquired using identical exposure settings for signal threshold determination and balancing. Representative images were flipped and/or cropped with contrast adjustments to highlight biological relevance using ImageJ (National Institutes of Health).

To quantify acetylated tubulin integrated density, the injected and uninjected areas of every second consecutive spinal cord section were manually selected using the freehand selection tool and measured in ImageJ. At least eight transverse 20 µm sections were analyzed per animal, spanning the length of more than 300 µm in developing spinal cords. The integrated density for the injected and uninjected sides was averaged separately for every animal (mean integrated density). Then, a ratio for these averages – the integrated density ratio – was calculated for every animal. To determine Sox2/3-positive cell numbers, individual cells from at least eight alternating

serial sections were counted on the injected and uninjected sides. For every animal, the ratio for the average number of cells in the injected to uninjected sides was calculated. The percentage of PH3+ cell nuclei out of all ependymal DAPI-positive nuclei was calculated for quantification of cell division on the injected and uninjected sides. For every animal, the ratio of the percentage of PH3+ cell nuclei as well as the ratio of DAPI+ nuclei in the injected to uninjected sides was calculated. After SCI, at least five horizontal spinal cord 20 µm sections were obtained per animal and condition. The length between the regenerating stumps was calculated using ImageJ, and averaged for every animal. To quantify Marcks/Marcksl1 integrated density, the entire area of every second consecutive spinal cord section was measured in ImageJ. Six to ten horizontal 20 µm sections were analyzed per animal, spanning 300 µm across the spinal cord and extending for 440 µm along the anteroposterior axis across the injury site and including the rostral and caudal stumps (i.e., the area depicted in each of the panels of *Figure 5*, *Figure 5—figure supplement 1*), seeking to include all cell types within this area during the injury response. Total cell counts from DAPI, EdU, and Sox2 images were also determined using ImageJ. DAPI-stained nuclei and EdU+ nuclei were counted in at least five horizontal sections per animal. Within each section, the entire population of cells in the spinal cord in a window extending 330 µm across the spinal cord and extending for 440 µm along the anteroposterior axis across the injury site and including the rostral and caudal stumps was analyzed (i.e., the area depicted in each of the panels of *Figure 7*). After binary conversion and particle analysis in ImageJ, the percentage of EdU+ or Sox2+ cells over the entire nuclear cell count was calculated.

Several statistical tests were employed for this study using GraphPad Prism (San Diego, California). A one-way ANOVA was chosen to compare how a single variable, MOs or RNPs, could affect neurite formation, cell proliferation, numbers of nuclei, or levels of Marcks or Marcksl1. The mean of each condition was then compared using the Tukey test, which compares every mean with every other mean. For pharmacological treatments on developing embryos, a one-way ANOVA was also used with a Dunnett test to compare every mean to a control mean, which was the DMSO condition. A two-way ANOVA was used to compare how CRISPR-gene disruption and injury time points can affect gap closure length after SCI. Here, an uncorrected Fisher's least significant difference test was used for pairwise comparisons of two specific groups (uninjected and CRISPR) against each other at every time point. Additionally, the mean percentage of EdU+ and Sox2+ cells in CRISPR and uninjected animals were compared at different time points using a two-way ANOVA with Sidak's test to compare every mean independently. Results are represented in box plots or bar plots indicating the range between the 25th and 75th percentiles, with all data points shown. Single data points represent the average value of a given measurement per animal, with error bars representing minimum to maximum values.

## Surgery for SCI

Before surgery, stage 50 tadpoles were placed in 0.1% MS222 in 0.1× MBS for approximately 30 s until the reflex response to touch was absent. The animals were then immobilized in a plasticine-coated Petri dish and rinsed with transplantation solution (88 nM NaCl, 1 mM KCl, 1 mM $MgSO_4 \cdot 7H_2O$, 5 mM 4-(2-hydroxyethyl)-1-piperazineethanesulfonic acid (HEPES), 3 mM $CaCl_2$, 100 units penicillin/ml, 0.1 mg/ml streptomycin), removing any excess solution. Dorsal skin and adjacent muscle tissues from the mid-thoracic level were incised using fine forceps, exposing the spinal cord. After removing a small portion of the meninges, a tungsten wire was inserted below the spinal cord to lift it slightly, facilitating subsequent transection with a 25-gauge injection needle. For sham injury, only the skin and muscle tissues were incised. Following surgery, the tadpoles were placed in a transplantation buffer for 30 min before being transferred to 0.1× MBS supplemented with antibiotics (400 mg/l penicillin, 400 mg/l streptomycin, and 25 mg/l gentamicin) for the duration of the study. The animals were housed in individual wells in a six-well plate in 0.1× MBS and supplemented with antibiotics, and the media was changed twice daily for the first 3 days post-injury and once daily after that. Feeding the animals began 2 days post-surgery, consisting of Micron Growth Food (Sera, 00720) in 0.1× MBS. For pharmacological treatments, tadpoles were incubated post-surgery in DMSO, FIPI, and S1P, using the aforementioned concentrations, and solutions were changed once daily for 7 days. Tadpoles were terminally anesthetized during the appropriate time points by prolonged exposure (20 min) in 0.1% MS222. Following euthanasia, the tadpoles were rinsed twice for 5 min in 1× PBS and fixed in 4% PFA overnight at 4°C.

## Swimming recovery test

A quantitative recovery test was designed by recording and measuring vibration-induced swimming to analyze the recovery of tadpoles following SCI. For this, tadpoles were placed in a custom-made six-well plate attached to a 10-mm Pico Vibe vibration motor (Precision Microdrives), connected via USB to a Mega 2560 MCU Board (Arduino). The vibration's duration was programmed using Arduino's Integrated Development Environment. Tadpoles were placed in the recording chamber for a 1-min equilibration period and subsequently recorded for 90 s, during which they were subjected to six 3-s stimulation pulses at equal time intervals. After recording, total swimming distances were measured using EthoVision XT video tracking software (Noldus) and analyzed in GraphPad Prism.

## Acknowledgements

This study was funded by Research Ireland and the European Regional Development Fund (ERDF) under grant number 13/RC/2073_P2. The authors acknowledge the use of the facilities of the Centre for Microscopy and Imaging at the University of Galway, Galway, Ireland, a facility that is co-funded by the Irish Government's Programme for Research in Third Level Institutions, Cycles 4 and 5, National Development Plan 2007–2013. During the course of this work MEA held a Hardiman fellowship from University of Galway and a Government of Ireland PhD Scholarship from the Irish Research Council (GOIPG/2018/500).

## Additional information

### Funding

| Funder | Grant reference number | Author |
| --- | --- | --- |
| Research Ireland | 13/RC/2073_P2 | Abhay Pandit |
| European Regional Development Fund | 13/RC/2073_P2 | Abhay Pandit |
| Irish Research Council for Science, Engineering and Technology | GOIPG/2018/500 | Mohamed El Amri |

The funders had no role in study design, data collection, and interpretation, or the decision to submit the work for publication.

### Author contributions

Mohamed El Amri, Investigation, Writing – original draft, Writing – review and editing; Abhay Pandit, Conceptualization, Supervision, Funding acquisition, Writing – review and editing; Gerhard Schlosser, Conceptualization, Supervision, Writing – original draft, Writing – review and editing

### Author ORCIDs

Mohamed El Amri https://orcid.org/0000-0002-7030-3192
Gerhard Schlosser https://orcid.org/0000-0002-1300-1331

### Ethics

All experiments were done in accordance with Irish and European legislation under licenses AE19125/P079 and AE19125/P098 from the Health Products Regulatory Authority (HPRA).

### Decision letter and Author response

Decision letter https://doi.org/10.7554/eLife.98277.sa1
Author response https://doi.org/10.7554/eLife.98277.sa2

## Additional files

### Supplementary files

• MDAR checklist

## Data availability

All data are made available as source data files (gel images) or by deposition into the Dryad database under https://doi.org/10.5061/dryad.b2rbnzsqm (all other data).

The following dataset was generated:

| Author(s) | Year | Dataset title | Dataset URL | Database and Identifier |
|---|---|---|---|---|
| Schlosser G, El Amri M, Pandit A | 2024 | Data from: Marcks and Marcks-like 1 proteins promote spinal cord development and regeneration in Xenopus | https://doi.org/10.5061/dryad.b2rbnzsqm | Dryad Digital Repository, 10.5061/dryad.b2rbnzsqm |

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

# Appendix 1

## Appendix 1—key resources table

| Reagent type (species) or resource | Designation | Source or reference | Identifiers | Additional information |
|---|---|---|---|---|
| Strain, strain background (*Xenopus laevis*) | *X. laevis*, Wild Type | 1 Corp, Michigan, USA | | |
| Sequence-based reagent | *marcksl1.L MO* | Gene Tools, LLC | Morpholinos | GGATTCTATACTACCCATTTTCCGC |
| Sequence-based reagent | *marcksl1.S MO* | Gene Tools, LLC; *Zhao et al., 2001* | Morpholinos | GGACTCTACGCTACCCATTGTGACT |
| Sequence-based reagent | *marcks MO* | Gene Tools, LLC; *Iioka et al., 2004* | Morpholinos | GGTCTTGGAGAATTGGGCTCCCATT |
| Sequence-based reagent | *Control MO* | Gene Tools, LLC | Morpholinos | CCTCTTACCTCAGTTACAATTTATA |
| Recombinant DNA reagent | marcks_OGa33947C_pET-23a(+) | GenScript | Plasmid | Clone ID: OGa33947C |
| Recombinant DNA reagent | marcksl1_OXa00945C_pET-23a(+) | GenScript | Plasmid | Clone ID: OXa00945C |
| Recombinant DNA reagent | marcks.S_ BC041207_ pCMV-Sport6 | Source Bioscience | Plasmid | Clone accession: BC041207 |
| Recombinant DNA reagent | Marcksl1.L_ BC084888_ pCMV-Sport6.ccdb | Source Bioscience | Plasmid | Clone accession: BC084888 |
| Recombinant DNA reagent | Marcksl1.S_ BC157454_ pCMV-Sport6 | Source Bioscience | Plasmid | Clone accession: BC157454 |
| Recombinant DNA reagent | memGFP-pCS2+ | *Wallingford et al., 2000* | Plasmid | Membrane GFP |
| Recombinant DNA reagent | pCMTEGFP | Kindly provided by Doris Wedlich | Plasmid | c-Myc-GFP |
| Sequence-based reagent | marcks.L/S | This paper | Primers for sgRNA synthesis | CTAGCTAATACGACTCAC TATAGGCTGCGGTCTTGGA GAATTGTTTTAGAGCTAGAAATAGCAAG |
| Sequence-based reagent | *marcksl1.L/S* | This paper | Primers for sgRNA synthesis | CTAGCTAATACGACTCACTATA GGTTGGGGTCCCCGTT GGCGGTTTTAGAGCTAGAAATAGCAAG |
| Sequence-based reagent | *slc45a2* | *DeLay et al., 2018* | Primers for sgRNA synthesis | CTAGCTAATACGACTCACTATA GGTTACATAGGCTGCCT CCAGTTTTAGAGCTAGAAATAGCAAG |
| Sequence-based reagent | *Universal reverse* | *DeLay et al., 2018* | Primers for sgRNA synthesis | AAAAGCACCGACTCGGTGCC ACTTTTTCAAGTTGATAA CGGACTAGCCTTATTTTAACTTG CTATTTCTAGCTCTAAAAC |
| Sequence-based reagent | *marcks.L_F* | This paper | PCR primers | ATCACCTGATGGACGCATGG |
| Sequence-based reagent | *marcks.L_R* | This paper | PCR primers | CCCCCACATCTAAAGCGGAG |
| Sequence-based reagent | *marcks.L* | This paper | Sequencing primers | CCCATGCTGTCTGTCTTTGA |
| Sequence-based reagent | *marcks.S_F* | This paper | PCR primers | AGTGTCATGAATCAGCGGGG |
| Sequence-based reagent | *marcks.S_R* | This paper | PCR primers | GTGTGTCTATTAGCGGCGGA |
| Sequence-based reagent | *marcks.S* | This paper | Sequencing primers | GTGTGTCTATTAGCGGCGGA |
| Sequence-based reagent | *marcksl1l1.L_F* | This paper | PCR primers | GCTAGGAAGAAGCGAGTCCC |
| Sequence-based reagent | *marcksl1.L_R* | This paper | PCR primers | ACCCGTTAACATGAGCAGCA |
| Sequence-based reagent | *marcksl1.L* | This paper | Sequencing primers | TCTTCTTCTGGCGCCTGC |
| Sequence-based reagent | *marcksl1l1.S_F* | This paper | PCR primers | AGAAAGAAGTGAGCCTAGAGTGATT |
| Sequence-based reagent | *marcksl1.S_R* | This paper | PCR primers | ATCCCTCCAAGGGTGACAGG |
| Sequence-based reagent | *marcksl1.S* | This paper | Sequencing primers | AGTTTAGGGAGGCAGGGTTG |

*Appendix 1 Continued on next page*

*Appendix 1 Continued*

| Reagent type (species) or resource | Designation | Source or reference | Identifiers | Additional information |
|---|---|---|---|---|
| Sequence-based reagent | *slc45a2.L_F* | **DeLay et al., 2018** | PCR primers | GTTCCCTTCGCTCATACAATG |
| Sequence-based reagent | *slc45a2.L_R* | **DeLay et al., 2018** | PCR primers | GCCAGAAAGGGGTTTATTGC |
| Sequence-based reagent | *slc45a2.L* | **DeLay et al., 2018** | Sequencing primers | GTTCCCTTCGCTCATACAATG |
| Strain, strain background (*Escherichia coli*) | XL1-Blue | Stratagene | 200249 | Chemically Competent Cells |
| Antibody | Anti-Tubulin, Acetylated (Mouse monoclonal) | Sigma | T6793 | IHC 1:1000 |
| Antibody | Anti-Phospho-Histone H3 (Rabbit Polyclonal) | Merck | 06-570 | IHC 1:200 |
| Antibody | Anti-GFP (Rabbit Polyclonal) | Abcam | ab290 | IHC 1:250 |
| Antibody | Anti-GFP (Mouse Monoclonal) | Abcam | ab1218 | IHC 1:250 |
| Antibody | Anti-c-Myc (Mouse Monoclonal) | Developmental Studies Hybridoma Bank | 9E 10 | IHC 1:200 |
| Antibody | Anti-c-Myc (Rabbit Polyclonal) | Merck | PLA0001 | IHC 1:200 |
| Antibody | Anti-6X-His-Tag (Chicken Polyclonal, FITC) | Abcam | ab3554 | IHC 1:100 |
| Antibody | Anti-T7-Tag (Goat Polyclonal, FITC) | Invitrogen | PA1-32390 | IHC 1:200 |
| Antibody | *Anti-X. laevis* MARCKS (Rabbit Polyclonal) | Genscript | Custom-made | IHC 1:200 |
| Antibody | *Anti-X. laevis* MARCKSL1 (Rabbit Polyclonal) | Genscript | Custom-made | IHC 1:100 |
| Antibody | Anti-Sox2 (Rabbit Monoclonal) | Cell Signalling Technology | D6D9 | IHC 1:100 |
| Antibody | Anti-Sox3 (Rabbit Polyclonal) | **Zhang et al., 2004** | Custom-made | IHC 1:100 |
| Chemical compound, drug | Phorbol 12-myristate 13-acetate (PMA) | TOCRIS | 1201 | 0.01 µM |
| Chemical compound, drug | 5-Fluoro-2-indolyl des-chlorohalopemide (FIPI) | TOCRIS | 3600 | 10 µM |
| Chemical compound, drug | Go 6983 | TOCRIS | 2285 | 10 µM |
| Chemical compound, drug | *N*-Methyllidocaine iodide (NMI) | TOCRIS | 1042 | 10 µM |
| Chemical compound, drug | Sphingosine-1-phosphate (S1P) | TOCRIS | 1370 | 1 µM |
| Chemical compound, drug | U 73122 | TOCRIS | 1268 | 10 µM |
| Chemical compound, drug | *m-3M3FBS* | TOCRIS | 1941 | 10 µM |
| Chemical compound, drug | ISA-2011B | MedChemExpress | HY-16937 | 10 µM |
| Chemical compound, drug | DMSO | Sigma-Aldrich | D2650 | 2% |
| Commercial assay or kit | Q5 site-directed mutagenesis kit | New England Biolabs | E0554S | |
| Commercial assay or kit | Ambion mMessage mMachineTM T7/SP6 Transcription Kit | Invitrogen | AM1340/44 | |
| commercial assay or kit | Poly(A) Tailing Kit | Invitrogen | AM1350 | |
| Commercial assay or kit | MEGAshortscript T7 Transcription Kit | Invitrogen | AM1354 | |

*Appendix 1 Continued*

| Reagent type (species) or resource | Designation | Source or reference | Identifiers | Additional information |
|---|---|---|---|---|
| Commercial assay or kit | Q5 High-Fidelity DNA Polymerase | New England Biolabs | M0491S | |
| Commercial assay or kit | TNT SP6 Coupled Wheat Germ Extract System | Promega | L4130 | |
| Commercial assay or kit | Click-iT EdU Cell Proliferation Kit for Imaging, Alexa Fluor 488 dye | Invitrogen | C10337 | |
| Commercial assay or kit | Roche Digoxigenin RNA Labeling Kit | Sigma-Aldrich | 11175025910 | |
| Commercial assay or kit | QIAquick PCR Purification Kit | QIAGEN | | |
| Commercial assay or kit | EmeraldAmp GT PCR Master Mix | Takara Bio | RR310A | |
| Commercial assay or kit | Wizard Genomic DNA Purification Kit | Promega | A1120 | |
| Other | TrueCut Cas9 Protein v2 | Invitrogen | A36497 | 0.2 ng/nl; see Materials and methods, **Microinjections** |
| Other | Streptavidin Alkaline Phosphatase | Promega | V5591 | 1:5000; see Materials and methods, **MO** |
| Other | DAPI stain | Invitrogen | D1306 | 100 ng/µl; see Materials and methods, **Histochemistry** |
| Software | ImageJ | PMID:22930834; https://imagej.nih.gov/ij/ | RRID:SCR_003070 | Software for image viewing and processing |
| Software | NEBaseChanger | New England Biolabs; https://nebasechanger.neb.com | | Cloning website |
| Software | Prism | GraphPad; https://www.graphpad.com/updates/prism-900-release-notes | RRID:SCR_002798 | Software for statistical analysis |
| Software | Synthego ICE | Synthego; https://www.synthego.com/guide/how-to-use-crispr/ice-analysis-guide | RRID:SCR_024508 | Website for CRISPR editing assessment |
| Software | CRISPRscan | CRISPRscan; https://www.crisprscan.org/ | RRID:SCR_023777 | Website for sgRNA design |
| Software | InDelphi | InDelphi; https://indelphi.giffordlab.mit.edu/ | | Website for predicting CRISPR outcomes |
| Software | GGGenome | GGGenome; https://gggenome.dbcls.jp/ | | Website for visualizing CRISPR off-targets |
| Software | EthoVision XT | EthoVision; https://www.noldus.com/ethovision-xt | RRID:SCR_000441 | Software for animal video tracking |

