## [Editor Report]

This important work addresses the role of Marcks/Markcksl during spinal cord development and regeneration. The study is exceptional in combining molecular approaches to understand the mechanisms of tissue regeneration with behavioural assays, which is not commonly employed in the field. The data presented is convincing and comprehensive, using many complementary methodologies.

---

## [Decision Letter]

**Decision letter after peer review:**

Thank you for submitting your article "Marcks and Marcks-like 1 proteins promote spinal cord development and regeneration in *Xenopus*" for consideration by *eLife*. Your article has been reviewed by 3 peer reviewers, and the evaluation has been overseen by a Reviewing Editor and Kathryn Cheah as the Senior Editor.

Essential revisions:

1) The reviewers have raised several concerns about potential overinterpretations in their conclusions from their data and areas where the text could be improved for clarity of the text. Please read carefully through the extensive list of the reviewers' comments and modify the text and figures as suggested by the three reviewers (see details below).

2) Please comment and modify the text (and provide data if available) that addresses the potential spinal cord rescue experiments in froglets, whether via misexpression or using drugs (see comment 3 by Reviewer 2).

3) Please comment and modify the text (and provide data if available) that addresses the potential cell-autonomous vs non-cell-autonomous mechanisms involved downstream of Marcks/Markcksl, (see comment 4 by Reviewer 2).

*Reviewer #1 (Recommendations for the authors):*

Whilst I suggest some further experiments below, most of my comments can be addressed either by being more careful in the wording of the conclusions from the data presented in this manuscript and / or by re-analysing / changing the presentation of existing data.

– MOs against marcks and marcksl1 have been used before in *Xenopus laevis* as mentioned by the authors in the methods. The phenotype of these knockdown experiments were quite severe (failure to gastrulate and shorten axis respectively Iioka et al. 2004, Zhao et al. 2001). As far as I can tell, the MOs were injected at a higher concentration than in this study, however the phenotype does make sense in terms of what we know of the function of these proteins and their peak of expression during gastrulation. Did the authors observe the same phenotype when the MOs are injected at a similar concentration? At the very least, previous studies describing knockdown phenotype should be discussed.

– Given the published papers cited above and the expression pattern (mRNAs and proteins) described by the authors, there is a possibility that the defects observed are not directly due to the absence of the Marcks in neural progenitors / neurons but are secondary defects of a wider phenotype (i.e. mesoderm formation). Have the authors look at other tissues (muscle, notochord) to ensure that they were formed properly. It is interesting that knockouts in mice mainly lead to neuronal defects despite being expressed in many other tissues (even if I have not seen a paper describing a double knockout of marcks / marcksl1), a point that could be discussed in this manuscript.

– I would not describe F0 Crispr-injected tadpoles as "knockouts", the term F0-Crispants is more appropriate otherwise it becomes difficult to distinguish from a "real" knockout strain (which I am not suggesting to generate, given the complexity of knocking out 4 different genes).

– The authors went on to use F0-Crispants to perform spinal cord transection at NF50. Given the early phenotype observed in MO / Crispr-Cas9 injected embryos, it is not easy to imagine that NF50 would have a functional spinal cord. This is discussed by the activation of Marcks-independent mechanisms, but could the authors expand on what may be happening mechanistically? An alternative explanation would be that only the tadpoles that can overcome the Crispr injection were selected, and therefore only a hypomorph phenotype is observed. Knowing the survival rate of Crispr-injected tadpoles vs controls at NF50 would be a start to address this issue.

– The description of pattern of expression of the mRNA is confusing to me. First some of this was done elsewhere (Gawantka et al. 1998; Zhao et al. 2001) and it should be acknowledged. Given the sequence similarities between marcks.S and L, I would label the in situs in Figure 1 / Figure S1 marcks or marcks.S/L. In the supplementary figures (S2/3/4) it would help non-*Xenopus* specialists to have a schematic of the embryos showing where the sections are on the AP axis. In particular when comparing marcks1l.L and S, the sections in S3A/B and S4A/B seem to be at very different levels along the AP axis. Finally, are any of the marcks expressed in the progenitor cells in the spinal cord, I cannot really be sure from the picture provided (in particular for marcksl1). The IF experiments could clarify this issue, but as they are presented, it is difficult to establish in which cell types are the Marcks expressed.

– Concluding from the anti-Acetylated Tubulin staining that the phenotype is a decrease in the number / length of neurites is not appropriate. Anti-acetylated Tubulin staining will also label the cell bodies of neurons and a reduction of neurons will give rise to fewer neurites / axon without their growth being affected. Given the decrease of progenitor proliferation, an alternative explanation would be an overall decrease of the number of neurons. Therefore the authors need to quantify the proportion of neurons in the wildtype and morphant/crispant spinal cords (using a neuronal marker such as Myt1). Labelling of individual neurons by injecting a low dose of a plasmid encoding mb-GFP in the blastomere giving rise to the spinal cord would also help characterising the phenotype (see Panagiotaki et al. 2010). Finally, did the authors notice any defects in motor / sensory axons that project outside the spinal cord (can be visualised both on sections and whole mount)?

– The authors observed a decrease in proliferation in the morphants/cripants during development and regeneration. Whilst at NF34-36, this is most probably due to a decrease in proliferation of *Sox2*/3+ cells, this is less clear after injury. As noted by the authors, multiple cell types invade the injury site, double labelling of EdU and *Sox2* would help understand which cell types are affected by the reduction in expression of Marcks and Marcksl1.

– Using chemical inhibitors, the authors argue that the mechanisms by which Marcks / Marcksl1 exert their roles is by promoting production of PIP2, leading to the activation of PLD. Conversely, PKC-dependant phosphorylation inhibits Marcks activity. I am not an expert on phospholipid signalling but it seems to me that S1P is far from being a specific PLD activator. Similarly, PMA is not a specific PKC activator, I cannot comment on NMI/ISA-2011B. Therefore the conclusion from these experiments needs to be carefully worded as to not convey this message (and reference to S1P as being a specific activator of PLD should be removed, i.e. l 257 / 494 / 1188 / 1200, etc). If the role of Marcks is to activate PLD then FIPI treatments on Crispr-injected embryos should not have an effect (Figure 9) when this is clearly not the case, indicating that either some Marcks activity is still present in the 4M Crispr (and therefore it is not a knockout) or some Marcks-independent activity of PLD remains. For this set of experiments, I am not suggesting further experiments but to clarify the text, presenting clearly the caveats of the approach taken.

– This is confounded by the way the results are presented: in Figure 4A/B, S1P treatment leads to a ratio of ~1 when comparing injected vs uninjected side. Given that the chemical treatment is applied to the whole embryo, there are 2 ways to get to a ratio of 1: one is by rescuing the phenotype on the injected side, the other is by decreasing the AcTub staining / %PH3 staining on the uninjected side. As the data are currently presented, I cannot distinguish between these possibilities. The authors need to show absolute numbers rather than the ratio of injected vs uninjected. The same applies to the rest of Figure 4.

– Injections: can the authors be more precise about what was injected where? Injection at the 2-cell stage will not target the dorsal ectoderm (l 614). Were all the MO experiments done by injecting 1 blastomere at the 2-cell stage? When injecting a single blastomere at 4/8 cells, which blastomere was targeted? Given that for some experiments only 1 blastomere at 4/8 cell stage is targeted (i.e. Figure 3), should only GFP+ve cells (the tracer) be taken into account when quantifying the phenotype. For example Figure 3C shows barely any cells in the spinal cord that are GFP+ve.

– in Figure 3 the authors argue that there is a 50% reduction in the number of PH3+ cells in MO/Crispr injected side vs uninjected side (Figure 3D/H) whereas the images consistently show no PH3+ cells in the injected side and 50% PH3+ cells in the rescue (3D). Can they show representative images and outline the area of the spinal cord.

– L397: what is the length of the cell cycle of progenitors after injury and the bio-availability of EdU? Thuret et al. 2015 calculated the bio-availability of BrdU at 2.5h and Pelzer et al. 2021 established the cell cycle length in the regenerating spinal cord at 50h (after tail amputation in tropicalis which is different to what is done here). A pulse of 16h is unlikely to label the entire population of proliferating cells. A double staining with PCNA could answer this question. Otherwise the text needs to be corrected (in the methods and the text).

– Whilst the upregulation of Marcks in the ependymal cell compartment is pretty clear (Figure 5), the data for Marcks1 (Figure S14) is much less convincing. Can the authors quantify (by PCR, WB or % of positive cells) these data?

*Reviewer #2 (Recommendations for the authors):*

The authors should consider the following points to improve the manuscript.

1) The concern about mosaicism of crispants (related to weakness #2),

As mentioned in the public review session, the usage of crispants in functional assay involves the issue of reproducibility. Their results would be further strengthened if the authors could show a similar phenotype with a gRNA that binds a different genomic sequence. Alternatively, the authors could include the other gRNA test results (total 18 gRNAs) in the manuscript, although they might not be as efficient as the ones used in the functional experiments.

2) The concern about the pharmacological rescue assay,

a) Due to the concern that was pointed out in the public review part (weakness #3, specificity), the authors might consider applying a few more compounds to the assay. Or the authors could deliver the candidate gene (PLD) by overexpression directly in the loss of function conditions. Alternatively, the authors should include the results of titration assays in the manuscript, since those results are particularly helpful for the readers that use similar model animals.

b) During SC regeneration, the authors tested PLD inhibitor (FIPI) only in the loss of function condition (Figure 9). Since there was no proper control for the inhibitor, the inhibitor needs to be tested in wild type condition as well. One interesting point here from the data (Figure 4 and Figure 9) would be that there is a difference between SC development and regeneration in terms of the response to FIPI in the loss of function condition. During SC development, FIPI treatment did not affect the number of proliferative cells but during regeneration it did. If the authors could include the proper control, it could be interesting to discuss the difference in the manuscript.

3) Rescue experiments of SC regeneration in froglet,

The authors mentioned Marcks and Marcksl1 are down-regulated in non-regenerative froglet (L54-57). It could be worth trying over expression Marcks and Marcksl1 during SC regeneration in the froglet stage and assess the number of the proliferative progenitor cells and measure gap distance after SC transection. It could be possible to deliver the genes to the injured site with electroporation or AAVs.

4) Cell autonomous or non-cell autonomous,

As the authors pointed out in the text (L481-484), whether the function comes from cell autonomous manner or non-cell autonomous manner would be an interesting question, especially in regenerative animals. Therefore, if the authors could successfully label knock-out cells specifically, then the knock-out cells from the neural plate can be transplanted in the neural plate of wild type embryos (embryonic transplantation). At a later stage, neurite outgrowth and progenitor cell proliferation can be analyzed in the same way on the knock-out cells specifically in wild type environments. Such types of experiments and vice versa can give more depth in the manuscript.

5) Issues on immunostaining images,

a) Figure 1. K, L, M: for broader readers, it would be difficult to see where the SC in the section is. It is better to use some SC markers to label the cells or simply enclose SC with a dash line in the images.

b) Figure 1. K-P: at L131, the authors mentioned Marcks is associated with the outer membrane. However, it is not a convincing image. The author needs to provide even higher resolution images or need different approaches, for example, immuno-electron microscopy.

c) Figure 5. B, E, H, K: It seems Marcks was upregulated during SC regeneration, however it is not clear. If the authors would like to argue the upregulation, it would be better to quantify the signal intensity or number of positive cells per area or

perform quantitative RT-PCR to support the idea.

*Reviewer #3 (Recommendations for the authors):*

1) Line 116: please replace "strong" with "expressed" for markscl signal in ISH, as ISH is not quantitative.

2) Figure 1 memGFP experiments are effective. Please consider presenting confocal images with single colors side by side to clarify overlaps, possibly in a supplementary figure.

3) Throughout the manuscript, please refer to CRISPR-injected F0 animals as "crispants" aligning with previous literature terminology for mosaic animals (as it has been carefully noted already).

4) Normally, it would have been necessary to use non-targeting guides for controls, instead of/in addition to uninjected controls. It was a bit surprising to see the authors didn't do this, considering their well-thought experiments throughout the manuscript. However, this work already demonstrates convincing data demonstrating the role of marcks. So, I am noting this comment for their future research endeavours.

5) In the introduction, the authors mention the loss of regeneration capacity in *Xenopus* during development. It is a missed opportunity to explore the role of marcks proteins in regeneration-incompetent tadpoles. This could begin by validating previous mass spec experiments with marcks antibodies and investigating if overexpression of marcks can induce spinal cord regeneration. However, these suggestions do not detract from the manuscript's current strength; they could be considered for future studies.

---

## [Author Response]

Essential revisions:1) The reviewers have raised several concerns about potential overinterpretations in their conclusions from their data and areas where the text could be improved for clarity of the text. Please read carefully through the extensive list of the reviewers' comments and modify the text and figures as suggested by the three reviewers (see details below).2) Please comment and modify the text (and provide data if available) that addresses the potential spinal cord rescue experiments in froglets, whether via misexpression or using drugs (see comment 3 by Reviewer 2).3) Please comment and modify the text (and provide data if available) that addresses the potential cell-autonomous vs non-cell-autonomous mechanisms involved downstream of Marcks/Markcksl, (see comment 4 by Reviewer 2).

We thank the reviewers for their many constructive comments and have incorporated these into our revised text as explained below making all essential revisions. We did tone down our conclusions, where requested and did add comments on the potential spinal cord rescue experiments in froglets as well as on the autonomous/non-autonomous role of Marcks/Marcksl1 (see responses to reviewers below for details). We also provided some additional data on pharmacological treatments in embryos. However, we did not do any additional experiments on spinal cord regeneration in tadpoles because the animal license for doing these experiments has expired.

Reviewer #1 (Recommendations for the authors):Whilst I suggest some further experiments below, most of my comments can be addressed either by being more careful in the wording of the conclusions from the data presented in this manuscript and / or by re-analysing / changing the presentation of existing data.

We thank the reviewer for the thoughtful comments. As explained below, we have incorporated most of them and feel that this has greatly improved the paper. However, for the reasons explained above, we did not do additional experiments.

– MOs against marcks and marcksl1 have been used before in *Xenopus laevis* as mentioned by the authors in the methods. The phenotype of these knockdown experiments were quite severe (failure to gastrulate and shorten axis respectively Iioka et al. 2004, Zhao et al. 2001). As far as I can tell, the MOs were injected at a higher concentration than in this study, however the phenotype does make sense in terms of what we know of the function of these proteins and their peak of expression during gastrulation. Did the authors observe the same phenotype when the MOs are injected at a similar concentration? At the very least, previous studies describing knockdown phenotype should be discussed.

To address this point, we added the following sentence:

“To minimize the gastrulation and axis defects previously observed after knockdown of *marcks* or *marcksl1* (Zhao et al. 2001; Iioka et al., 2004), we injected lower amounts of MOs (9 ng) than in previous studies (16 – 40 ng).”

– Given the published papers cited above and the expression pattern (mRNAs and proteins) described by the authors, there is a possibility that the defects observed are not directly due to the absence of the Marcks in neural progenitors / neurons but are secondary defects of a wider phenotype (i.e. mesoderm formation). Have the authors look at other tissues (muscle, notochord) to ensure that they were formed properly. It is interesting that knockouts in mice mainly lead to neuronal defects despite being expressed in many other tissues (even if I have not seen a paper describing a double knockout of marcks / marcksl1), a point that could be discussed in this manuscript.

We did not note any obvious deficiencies in somites or notochord but did not investigate this systematically so cannot make any specific statements. We already address the possibility that effects of Marcks may be direct or indirect in our discussion.

– I would not describe F0 Crispr-injected tadpoles as "knockouts", the term F0-Crispants is more appropriate otherwise it becomes difficult to distinguish from a "real" knockout strain (which I am not suggesting to generate, given the complexity of knocking out 4 different genes).

We reworded this throughout the paper.

– The authors went on to use F0-Crispants to perform spinal cord transection at NF50. Given the early phenotype observed in MO / Crispr-Cas9 injected embryos, it is not easy to imagine that NF50 would have a functional spinal cord. This is discussed by the activation of Marcks-independent mechanisms, but could the authors expand on what may be happening mechanistically? An alternative explanation would be that only the tadpoles that can overcome the Crispr injection were selected, and therefore only a hypomorph phenotype is observed. Knowing the survival rate of Crispr-injected tadpoles vs controls at NF50 would be a start to address this issue.

We prefer not to speculate, which Marcks-independent mechanisms may help these tadpoles to catch up with spinal cord development, since we cannot back this up with any evidence and it would not contribute to our understanding of the role of Marcks/Marcksl1. It is unlikely that we selectively picked hypomorphs since the survival rate of CRISPR injected tadpoles was not noticeable different from wild types. We did not do any counts, however, to quantify this.

– The description of pattern of expression of the mRNA is confusing to me. First some of this was done elsewhere (Gawantka et al. 1998; Zhao et al. 2001) and it should be acknowledged. Given the sequence similarities between marcks.S and L, I would label the in situs in Figure 1 / Figure S1 marcks or marcks.S/L. In the supplementary figures (S2/3/4) it would help non-*Xenopus* specialists to have a schematic of the embryos showing where the sections are on the AP axis. In particular when comparing marcks1l.L and S, the sections in S3A/B and S4A/B seem to be at very different levels along the AP axis. Finally, are any of the marcks expressed in the progenitor cells in the spinal cord, I cannot really be sure from the picture provided (in particular for marcksl1). The IF experiments could clarify this issue, but as they are presented, it is difficult to establish in which cell types are the Marcks expressed.

Thanks to the reviewer to pointing out the missing references, which we added. We also relabelled the images in Figure 1 as marcks.S/L as requested. For the sections we clarified in the legend the level of sections. These are at relatively comparable levels for all three Figures(forebrain, hindbrain, spinal cord) but in the embryo depicted in Figure S4 (now Figure 1- Figure Suppl. 4) the hindbrain seems to have been somewhat compressed laterally so that the fourth ventricle appears compressed and the thin roof of the fourth ventricle, which is clearly visible in Figures S2 and S3 (now Figure 1- Figure suppls 2 and 3) is folded up. In the immunostained sections shown in Figure 1 L,M,O,P we now also labelled the central canal to make it clear that Marcks and Marcksl1 both are also expressed in progenitor cells of the ventricular zone that directly contact the central canal (described in line 133-136 in the main text)..

– Concluding from the anti-Acetylated Tubulin staining that the phenotype is a decrease in the number / length of neurites is not appropriate. Anti-acetylated Tubulin staining will also label the cell bodies of neurons and a reduction of neurons will give rise to fewer neurites / axon without their growth being affected. Given the decrease of progenitor proliferation, an alternative explanation would be an overall decrease of the number of neurons. Therefore the authors need to quantify the proportion of neurons in the wildtype and morphant/crispant spinal cords (using a neuronal marker such as Myt1). Labelling of individual neurons by injecting a low dose of a plasmid encoding mb-GFP in the blastomere giving rise to the spinal cord would also help characterising the phenotype (see Panagiotaki et al. 2010). Finally, did the authors notice any defects in motor / sensory axons that project outside the spinal cord (can be visualised both on sections and whole mount)?

In our hands the acetylated tubulin antibody only labels neurites strongly. Cell bodies are not or only very weakly labelled and the staining is, consequently, concentrated in the marginal zone of the spinal cord, where the neurite tracts run. We are therefore confident that the significant changes in acetylated tubulin staining that we observe reflect changes in neurite density (we did not specifically analyse motor or sensory neurites). We agree with the reviewer that this does not necessarily reflect changes in axon outgrowth and we, therefore worded our description of the phenotype more carefully. However, even though proliferation is decreased after Marcks/Marcksl1, overall cell numbers are not significantly different between injected and uninjected sides (see new Figures N, O). Reduction in cell number can, therefore, not account for the reduction in acetylated tubulin staining and we added the following statement: “In spite of the reduction in progenitor proliferation after Marcks/Marcksl1 loss of function, overall cell numbers (as indicated by the numbers of DAPI+ nuclei) on the injected side were not yet significantly reduced in these embryos (Figure 3 N, O) indicating that the decrease in neurite staining described in the previous section cannot be attributed to a reduction in overall cell number.”

– The authors observed a decrease in proliferation in the morphants/cripants during development and regeneration. Whilst at NF34-36, this is most probably due to a decrease in proliferation of Sox2/3+ cells, this is less clear after injury. As noted by the authors, multiple cell types invade the injury site, double labelling of EdU and Sox2 would help understand which cell types are affected by the reduction in expression of Marcks and Marcksl1.

We agree but unfortunately will not be able to do additional experiments as explained above

– Using chemical inhibitors, the authors argue that the mechanisms by which Marcks / Marcksl1 exert their roles is by promoting production of PIP2, leading to the activation of PLD. Conversely, PKC-dependant phosphorylation inhibits Marcks activity. I am not an expert on phospholipid signalling but it seems to me that S1P is far from being a specific PLD activator. Similarly, PMA is not a specific PKC activator, I cannot comment on NMI/ISA-2011B. Therefore the conclusion from these experiments needs to be carefully worded as to not convey this message (and reference to S1P as being a specific activator of PLD should be removed, i.e. l 257 / 494 / 1188 / 1200, etc). If the role of Marcks is to activate PLD then FIPI treatments on Crispr-injected embryos should not have an effect (Figure 9) when this is clearly not the case, indicating that either some Marcks activity is still present in the 4M Crispr (and therefore it is not a knockout) or some Marcks-independent activity of PLD remains. For this set of experiments, I am not suggesting further experiments but to clarify the text, presenting clearly the caveats of the approach taken.

We thank the reviewer for pointing this out and have completely rewritten our presentation and discussion of these results accordingly. We also have added additional data using activators and inhibitors of PLC signalling.

– This is confounded by the way the results are presented: in Figure 4A/B, S1P treatment leads to a ratio of ~1 when comparing injected vs uninjected side. Given that the chemical treatment is applied to the whole embryo, there are 2 ways to get to a ratio of 1: one is by rescuing the phenotype on the injected side, the other is by decreasing the AcTub staining / %PH3 staining on the uninjected side. As the data are currently presented, I cannot distinguish between these possibilities. The authors need to show absolute numbers rather than the ratio of injected vs uninjected. The same applies to the rest of Figure 4.

We have added a new supplemental Figure (Figure 4—figure supplement 1) to show the effects of compounds on injected and uninjected sides separately in absolute numbers. We have also added a statement to this section of the results explaining why we think that the presentation as ratios (which we keep in Figure 4) is the most appropriate way to present these data: “Because pharmacological compounds may affect the spinal cord in multiple ways, not all of which may be Marcks/Marcksl1-dependent, we present our data here as ratios of acetylated tubulin or PH3 staining in injected/uninjected sides (Figure 4). This allows us to specifically determine, whether compounds are able to rescue deficiencies in Marcks/Marcksl1-depleted spinal cords independent of their potentially additional effects on spinal cord development. For completeness, the individual effects of pharmacological treatments on the injected and uninjected sides of embryos are shown in a supplemental figure (Figure 4—figure supplement 1).”

– Injections: can the authors be more precise about what was injected where? Injection at the 2-cell stage will not target the dorsal ectoderm (l 614). Were all the MO experiments done by injecting 1 blastomere at the 2-cell stage? When injecting a single blastomere at 4/8 cells, which blastomere was targeted? Given that for some experiments only 1 blastomere at 4/8 cell stage is targeted (i.e. Figure 3), should only GFP+ve cells (the tracer) be taken into account when quantifying the phenotype. For example Figure 3C shows barely any cells in the spinal cord that are GFP+ve.

We clarified throughout the text that dorsal blastomeres were targeted in all injections at 4-8 cell stages. The distribution and intensity of GFP is nevertheless a bit variable on the injected side. Due to the variable GFP intensity, counting only GFP+ve cells would require to make arbitrary decisions about thresholds. We therefore opted for the more conservative approach of just comparing effects on injected vs uninjected side (which should, if anything, give smaller effect sizes).

– in Figure 3 the authors argue that there is a 50% reduction in the number of PH3+ cells in MO/Crispr injected side vs uninjected side (Figure 3D/H) whereas the images consistently show no PH3+ cells in the injected side and 50% PH3+ cells in the rescue (3D). Can they show representative images and outline the area of the spinal cord.

Due to the sparse PH3 labeling most sections on the injected side indeed show no PH3 staining, so these sections are representative. The spinal cord has now been outlined in all Figures showing PH3 staining.

– L397: what is the length of the cell cycle of progenitors after injury and the bio-availability of EdU? Thuret et al. 2015 calculated the bio-availability of BrdU at 2.5h and Pelzer et al. 2021 established the cell cycle length in the regenerating spinal cord at 50h (after tail amputation in tropicalis which is different to what is done here). A pulse of 16h is unlikely to label the entire population of proliferating cells. A double staining with PCNA could answer this question. Otherwise the text needs to be corrected (in the methods and the text).

We do not know the length of the cell cycle in the regenerating spinal cord of *Xenopus laevis* and therefore have reworded our description in the methods and Results section, where we now no longer refer to saturated labeling.

– Whilst the upregulation of Marcks in the ependymal cell compartment is pretty clear (Figure 5), the data for Marcks1 (Figure S14) is much less convincing. Can the authors quantify (by PCR, WB or % of positive cells) these data?

We now quantified Marcks and Marcksl1 immunostaining after SCI and added this as new Figures 5V and Figure 5 —figure supplement Figure 1U. Significant upregulation of both Marcks and Marcksl1 was observed at day 10 postinjury.

Reviewer #2 (Recommendations for the authors):The authors should consider the following points to improve the manuscript.1) The concern about mosaicism of crispants (related to weakness #2),As mentioned in the public review session, the usage of crispants in functional assay involves the issue of reproducibility. Their results would be further strengthened if the authors could show a similar phenotype with a gRNA that binds a different genomic sequence. Alternatively, the authors could include the other gRNA test results (total 18 gRNAs) in the manuscript, although they might not be as efficient as the ones used in the functional experiments.

We only analysed the total 18gRNAs for knockout efficiencies but did not subject them to a phenotypic analysis, so we cannot present additional phenotypic data. However, we feel that our morpholino data already address this issue of reproducibility, since we are able to show that they generate the same phenotype.

2) The concern about the pharmacological rescue assay,a) Due to the concern that was pointed out in the public review part (weakness #3, specificity), the authors might consider applying a few more compounds to the assay. Or the authors could deliver the candidate gene (PLD) by overexpression directly in the loss of function conditions. Alternatively, the authors should include the results of titration assays in the manuscript, since those results are particularly helpful for the readers that use similar model animals.

As detailed in our responses to reviewer, we have significantly toned down and revised our discussion of pharmacological experiments to address these concerns and added additional data obtained with a PLC activator and inhibitor.

b) During SC regeneration, the authors tested PLD inhibitor (FIPI) only in the loss of function condition (Figure 9). Since there was no proper control for the inhibitor, the inhibitor needs to be tested in wild type condition as well. One interesting point here from the data (Figure 4 and Figure 9) would be that there is a difference between SC development and regeneration in terms of the response to FIPI in the loss of function condition. During SC development, FIPI treatment did not affect the number of proliferative cells but during regeneration it did. If the authors could include the proper control, it could be interesting to discuss the difference in the manuscript.

We thank the reviewer for this suggestion, but unfortunately, we cannot do any additional experiments and we therefore cannot address any potential difference of the effect of FIP between spinal cord development and regeneration in normal (not CRISPR injected) *Xenopus*. We wish to point out, however, that to check for the capacity of FIPI to modulate the effect of Marcks/Marksl1-depletion we compare the effects of FIPI (experimental condition) to DMSO (control condition) in both embryos and during SC regeneration in tadpoles. To highlight the differences of FIPI responses between embryos and tadpoles, we added the following sentence: “In distinction to embryos, where FIPI has no effect, the latter finding suggests that residual PLD activity must be present in CRISPR tadpoles either because some Marcks/Marcksl1-independent PLD activity remains or because Marcks/Marcksl1 have not completely been knocked out in the entire spinal cord”

3) Rescue experiments of SC regeneration in froglet,The authors mentioned Marcks and Marcksl1 are down-regulated in non-regenerative froglet (L54-57). It could be worth trying over expression Marcks and Marcksl1 during SC regeneration in the froglet stage and assess the number of the proliferative progenitor cells and measure gap distance after SC transection. It could be possible to deliver the genes to the injured site with electroporation or AAVs.

We completely agree with this and these experiments were indeed planned and we already had obtained ethical approval to go ahead with them. Unfortunately, the student working on the project ran out of time and our animal license to conduct these experiments has now expired, so they will have to be done in future studies. We have now added the following sentence to point this out: “Our findings that Marcks and Marcksl1 are required for proper regeneration of the spinal cord after injury suggest that the downregulation of both proteins in postmetamorphic froglets (Kshirsagar, 2020) may be one of the factors preventing spinal cord regeneration in adults. To test this hypothesis, future studies should determine, whether spinal cord regeneration in froglets can be experimentally promoted by overexpression of these proteins.”

4) Cell autonomous or non-cell autonomous,As the authors pointed out in the text (L481-484), whether the function comes from cell autonomous manner or non-cell autonomous manner would be an interesting question, especially in regenerative animals. Therefore, if the authors could successfully label knock-out cells specifically, then the knock-out cells from the neural plate can be transplanted in the neural plate of wild type embryos (embryonic transplantation). At a later stage, neurite outgrowth and progenitor cell proliferation can be analyzed in the same way on the knock-out cells specifically in wild type environments. Such types of experiments and vice versa can give more depth in the manuscript.

Again, we agree that these would be great experiments, but unfortunately these will have to be done in future studies for the reasons explained above. To highlight this we added the following sentence to the paragraph discussing autonomous vs. non –autonomous effects: “Moreover, transplantations of spinal cords between F0-CRISPants and wildtype embryos will allow to address, whether these proteins are required autonomously or non-autonomously for spinal cord regeneration.“

5) Issues on immunostaining images,a) Figure 1. K, L, M: for broader readers, it would be difficult to see where the SC in the section is. It is better to use some SC markers to label the cells or simply enclose SC with a dash line in the images.

We outlined the spinal cord with a dashed line for clarity

b) Figure 1. K-P: at L131, the authors mentioned Marcks is associated with the outer membrane. However, it is not a convincing image. The author needs to provide even higher resolution images or need different approaches, for example, immuno-electron microscopy.

For additional clarity, we added some additional labelling (central canal) and outlined the spinal cord in K and N with a dashed line. Yellow arrows in LM, O and P identify membranes, double-labeled for mem-GFP and Marcks or Marcksl1 immunostaining.

c) Figure 5. B, E, H, K: It seems Marcks was upregulated during SC regeneration, however it is not clear. If the authors would like to argue the upregulation, it would be better to quantify the signal intensity or number of positive cells per area orperform quantitative RT-PCR to support the idea.

As explained above, we now quantified Marcks and Marcksl1 immunostaining after SCI and added this as new Figures 5V and Figure 5—figure supplement 1U; this confirmed significant upregulation of both Marcks and Marcksl1 at day 10 postinjury.

Reviewer #3 (Recommendations for the authors):1) Line 116: please replace "strong" with "expressed" for markscl signal in ISH, as ISH is not quantitative.

Corrected

2) Figure 1 memGFP experiments are effective. Please consider presenting confocal images with single colors side by side to clarify overlaps, possibly in a supplementary figure.

We added Figure 1- Figure suppl. 4 as a supplementary Figure showing the separate channels (+DAPI for orientation)

3) Throughout the manuscript, please refer to CRISPR-injected F0 animals as "crispants" aligning with previous literature terminology for mosaic animals (as it has been carefully noted already).

Corrected

4) Normally, it would have been necessary to use non-targeting guides for controls, instead of/in addition to uninjected controls. It was a bit surprising to see the authors didn't do this, considering their well-thought experiments throughout the manuscript. However, this work already demonstrates convincing data demonstrating the role of marcks. So, I am noting this comment for their future research endeavours.

Thank you for pointing this out (we have done the controls with non-targeting guides for the embryo experiments but not after SCI).

5) In the introduction, the authors mention the loss of regeneration capacity in *Xenopus* during development. It is a missed opportunity to explore the role of marcks proteins in regeneration-incompetent tadpoles. This could begin by validating previous mass spec experiments with marcks antibodies and investigating if overexpression of marcks can induce spinal cord regeneration. However, these suggestions do not detract from the manuscript's current strength; they could be considered for future studies.

We agree wholeheartedly, but as explained above, we were unable to complete our planned experiments of overexpressing Marcks/Marcksl1in froglets due to time constraints.